# The F-actin bundler SWAP-70 promotes tumor metastasis

Chao-Yuan Chang[1], Glen Pearce[1], Viktoria Betaneli[1], Tatsiana Kapustsenka[1], Kamran Hosseini[2], Elisabeth Fischer-Friedrich[2], Denis Corbeil[3,4], Jana Karbanová[3,4], Anna Taubenberger[3,4], Björn Dahncke[1], Martina Rauner[5], Giulia Furesi[5], Sven Perner[6,7], Fabian Rost[8], Rolf Jessberger[1]

**Dynamic rearrangements of the F-actin cytoskeleton are a hallmark of tumor metastasis. Thus, proteins that govern F-actin rearrangements are of major interest for understanding metastasis and potential therapies. We hypothesized that the unique F-actin binding and bundling protein SWAP-70 contributes importantly to metastasis. Orthotopic, ectopic, and short-term tail vein injection mouse breast and lung cancer models revealed a strong positive dependence of lung and bone metastasis on SWAP-70. Breast cancer cell growth, migration, adhesion, and invasion assays revealed SWAP-70's key role in these metastasis-related cell features and the requirement for SWAP-70 to bind F-actin. Biophysical experiments showed that tumor cell stiffness and deformability are negatively modulated by SWAP-70. Together, we present a hitherto undescribed, unique F-actin modulator as an important contributor to tumor metastasis.**

## Introduction

By far, the largest fraction of cancer-related deaths occurs through metastasis, and thus, metastasis is the greatest challenge to cancer therapy. Although new approaches have been established, such as immune checkpoint or cell-based therapies, additional new therapies are required to solve this most pressing problem for human health. Key steps in metastasis include the dissociation of tumor cells from the primary tumor, the entry into and migration through blood or lymph vessels, the exit from vessels, entry into tissues, and finally homing into a suitable tissue area to grow a metastatic tumor unless the tumor already develops within vessels. This stepwise process, governed by multiple interactions, is reminiscent of Stephen Paget's "seed and soil" theory of 1889 (Paget, 1989). All these steps require dynamic rearrangements of the F-actin cytoskeleton of the tumor cell and thus proteins that govern F-actin dynamics.

Indeed, highly active F-actin dynamics has been designated a hallmark for aggressive and/or metastasizing tumor cells, which form a range of F-actin–based structures such as lamellipodia, filopodia, and invadopodia to facilitate migration and invasion (Verschueren et al, 1994; Muller et al, 2001; Olson & Sahai, 2009; Hanahan & Weinberg, 2011; Nurnberg et al, 2011; Gross, 2013; Steeg, 2016; Lambert et al, 2017; Mondal et al, 2021). In recent years, F-actin modulatory and accessory proteins increasingly attracted attention as targets in metastasis prevention and/or therapy (Nurnberg et al, 2014; Huang et al, 2015; Yin et al, 2019; Barik et al, 2022; Limaye et al, 2022).

Another proposed hallmark of metastasis is the epithelial-to-mesenchymal transition (EMT) and its reversal MET (Zavadil & Bottinger, 2005; Moustakas & Heldin, 2007; Kalluri & Weinberg, 2009). The concept suggests that by acquiring a more mesenchymal state, tumor cells are mobilized at the primary site, and when forming a metastatic tumor upon settlement in the secondary tissue, an epithelial phenotype is re-established. The EMT-MET processes may not be strictly required for metastasis. Intermediate states with only gradual changes and mixed epithelial and mesenchymal cell phenotypes are observed and may be sufficient depending on the tumor type and environment (Lou et al, 2008; Liu et al, 2019). It is well known that growth factors actively participate in tumorigenesis by activating complex signaling pathways, and regulating cancer cell proliferation, division, migration, and invasion (Sever & Brugge, 2015). Transforming growth factor-$\beta$ (TGF$\beta$) and EGF are two of the primary growth factors that promote invasion and metastasis of breast cancer tumors (Thiery, 2002; Kalluri & Weinberg, 2009; Wendt et al, 2010). TGF$\beta$ and EGF are known to induce cytoskeletal rearrangements, which promote cell motility (Boland et al, 1996; Malliri et al, 1998; Felkl et al, 2012; Gladilin et al, 2019).

We described switch-associated protein 70 (SWAP-70) as a unique F-actin binding and bundling protein (Chacon-Martinez et al, 2013; Betaneli & Jessberger, 2020), which is required for several F-actin–dependent processes. These include formation of lamellipodia in B lymphocytes and their migration in cell culture

---

[1]Institute for Physiological Chemistry, Medical Faculty Carl Gustav Carus, Technische Universität Dresden, Dresden, Germany   [2]Cluster of Excellence Physics of Life, Technische Universität Dresden, Dresden, Germany   [3]Biotechnology Center (BIOTEC) and Center for Molecular and Cellular Bioengineering, Dresden, Germany   [4]Medical Faculty Carl Gustav Carus, Technische Universität Dresden, Dresden, Germany   [5]Department of Medicine III and Center for Healthy Aging, Medical Faculty Carl Gustav Carus, Technische Universität Dresden, Dresden, Germany   [6]Institute of Pathology, University of Lübeck and University Hospital Schleswig-Holstein, Lübeck, Germany   [7]Institute of Pathology, Research Center Borstel, Leibniz Lung Center, Borstel, Germany   [8]DRESDEN-concept Genome Center, Technology Platform at the Center for Molecular and Cellular Bioengineering (CMCB), Technische Universität Dresden, Dresden, Germany

Correspondence: rolf.jessberger@tu-dresden.de

and in mice (Pearce et al, 2006; Pearce et al, 2011), migration of dendritic cells toward chemokines and sphingosine-1-phosphate in cell culture and mice (Ocana-Morgner et al, 2011), formation of F-actin rings in osteoclasts (Garbe et al, 2012; Roscher et al, 2016), and migration of mast cells (Sivalenka & Jessberger, 2004). The protein is also recruited to circular ruffles at cellular edges, a process involved in macro-pinocytosis (Oberbanscheidt et al, 2007), and acts in tethering peripheral actin to phagosomes (Baranov et al, 2016). Treatment of tumor cells with the motility- and metastasis-modulating drug 12(S)-HETE caused SWAP-70 translocation to membrane ruffle-like structures where the protein dimerizes, binds, and bundles actin (Betaneli & Jessberger, 2020). These data strongly suggest a key role of SWAP-70 in promoting cell motility and invasion, processes that are required for tumor cells to metastasize. In addition, SWAP-70 most likely through actin-dependent control of receptor function supports cell adhesion (Sivalenka & Jessberger, 2004; Chopin et al, 2010; Ripich & Jessberger, 2011) and thus another process that is instrumental for metastatic tumor cells to reach and home into target organs. Among the prominent structural features of SWAP-70 are a pleckstrin homology (PH) domain enabling the protein to bind to phosphatidylinositol (3,4,5)-triphosphate (PIP$_3$) and more weakly to PI(3,4)P$_2$ at the cytoplasmic membrane (Shinohara et al, 2002; Wakamatsu et al, 2006; Kriplani et al, 2019) and a C-terminal F-actin binding domain (Ihara et al, 2006; Chacon-Martinez et al, 2013). Other domains and motifs include a putative EF-hand, a Dbl homology domain, and a dimerization domain located just upstream of the actin binding domain, which allows the protein to bundle actin filaments. There is no other protein in the vertebrate genome that shares the same domain arrangement; the closest other protein DEF6/IBP lacks the F-actin domain (Mavrakis et al, 2004). Studies also revealed that upon EGF stimulation, SWAP-70 translocated to the cell membrane and induced membrane ruffles and lamellipodia in several cell types (Shinohara et al, 2002; Fukui & Ihara, 2010; Chacon-Martinez et al, 2013). This suggests a link between these metastasis-relevant growth factors, cytoskeletal rearrangements, and SWAP-70.

Although clinical cancer databases do not present SWAP-70 as a major cancer-associated factor, SWAP-70 was experimentally implicated in a few cancer-related settings. The expression of SWAP-70 correlated with the severity of glioblastoma and was shown to affect in vitro extracellular matrix degradation and invasion (Seol et al, 2009; Dong et al, 2022), as well as anchorage-independent growth of tumor cell lines (Murugan et al, 2008; Shu et al, 2013). However, neither studies on the role of SWAP-70 in tumor metastasis in vivo were reported, nor were potential mechanisms described.

We hypothesized that SWAP-70 supports tumor cell metastasis through controlling F-actin dynamics, tested this hypothesis in a murine breast cancer setting, and revealed biophysical and molecular properties of SWAP-70 linked to this role.

# Results

### SWAP-70 promotes metastasis in vivo

To identify the role of SWAP-70 in mouse breast cancer metastasis, we generated SWAP-70–deficient (KO) cell lines using the CRISPR/

Cas9 system in the BALB/c-derived mouse breast cancer cell line 4T1, which resembles human metastatic triple-negative breast cancer cells (Dexter et al, 1978; Schrors et al, 2020). To minimize putative off-target effects, two distinct guide RNAs were designed to generate two KO cell lines, called G1KO and G3KO. At 48 h after transfection with the Cas9-GFP– and guide RNA–expressing plasmid, single GFP-positive cells were isolated by FACS. Cells transfected with a plasmid expressing only Cas9-GFP were prepared in the same manner to serve as a control (subsequently referred to as Ctrl). Clones were screened for SWAP-70 deficiency by immunoblotting (Fig S1A). To avoid a potential bias because of the use of single clones and to mimic some tumor cell heterogeneity, five clones from each, either Ctrl, G1KO, or G3KO, were mixed using equal cell numbers for all experiments. Each pool was orthotopically injected into the fourth mammary fat pad of female BALB/c mice to recapitulate metastasis from a primary tumor in vivo. Primary tumor growth was measured weekly, and the mice were euthanized upon occurrence of (1) body weight loss of >10%, (2) ulceration on the skin at the site of the primary tumor, or (3) deterioration of the health status (Fig 1A).

Primary tumors generated by Ctrl or G1KO cells grew at a similar rate, whereas G3KO cell–generated tumors grew to about half the volume (Fig 1B). It had been reported that tumors formed by 4T1 cells metastasize to multiple sites, such as lung, liver, kidney, and bones (Lelekakis et al, 1999; Pulaski & Ostrand-Rosenberg, 2001) with lung metastasis appearing first, followed later by bone metastasis (Tao et al, 2008). Therefore, we collected lungs and bones to examine metastasis as they are prominent metastatic sites and represent early- and late-stage metastasis in 4T1-derived tumors. We counted the visible metastatic nodules (referred to as macro-metastases) on the lungs defined as macro-metastasis based on images taken on a dissecting microscope. Mice bearing Ctrl cell–generated primary tumors showed 10-fold more lung macro-metastases compared with mice that carried primary tumors generated from either of the two KO cell pools, which often carried no macro-metastasis (Fig 1C). The difference in the primary tumor growth rate did not translate into a metastasis difference between G1KO and G3KO. The strong difference between Ctrl and KO cells was confirmed by lung histology through hematoxylin and eosin (H&E) staining of 7 Ctrl-, 8 G1KO-, and 7 G3KO-injected mice, revealing a 2.4-fold difference between Ctrl and G1KO, a 3.1-fold difference between Ctrl and G3KO, and no significant difference between G1KO and G3KO (Fig 1D). The average size of lung metastatic nodules was not different between the groups indicating that proliferation of metastatic cells itself was not compromised by the absence of SWAP-70 (area: Ctrl, 206.6 ± 138.2 mm$^2$; G1KO and G3KO taken together because of few metastases, 172.2 ± 183.7 mm$^2$; $t$ test, Ctrl versus. KO, $P = 0.35$). For analysis of bone metastasis, both hind leg bones from each mouse were collected for micro-computed tomography ($\mu$CT) scanning to observe osteolytic bone lesions. Three-dimensional reconstructive imaging allowed us to classify bone lesions into three levels of increasing severity (Fig 1E), whereas each sample was scored in a blinded manner. 26 legs from each group were analyzed. Level III osteolytic lesions were observed in 12 legs from Ctrl mice, whereas only 5 G1KO legs and no G3KO legs showed such osteolytic lesions. Histological examination of the bones revealed a 1.9-fold difference between Ctrl and G1KO and a

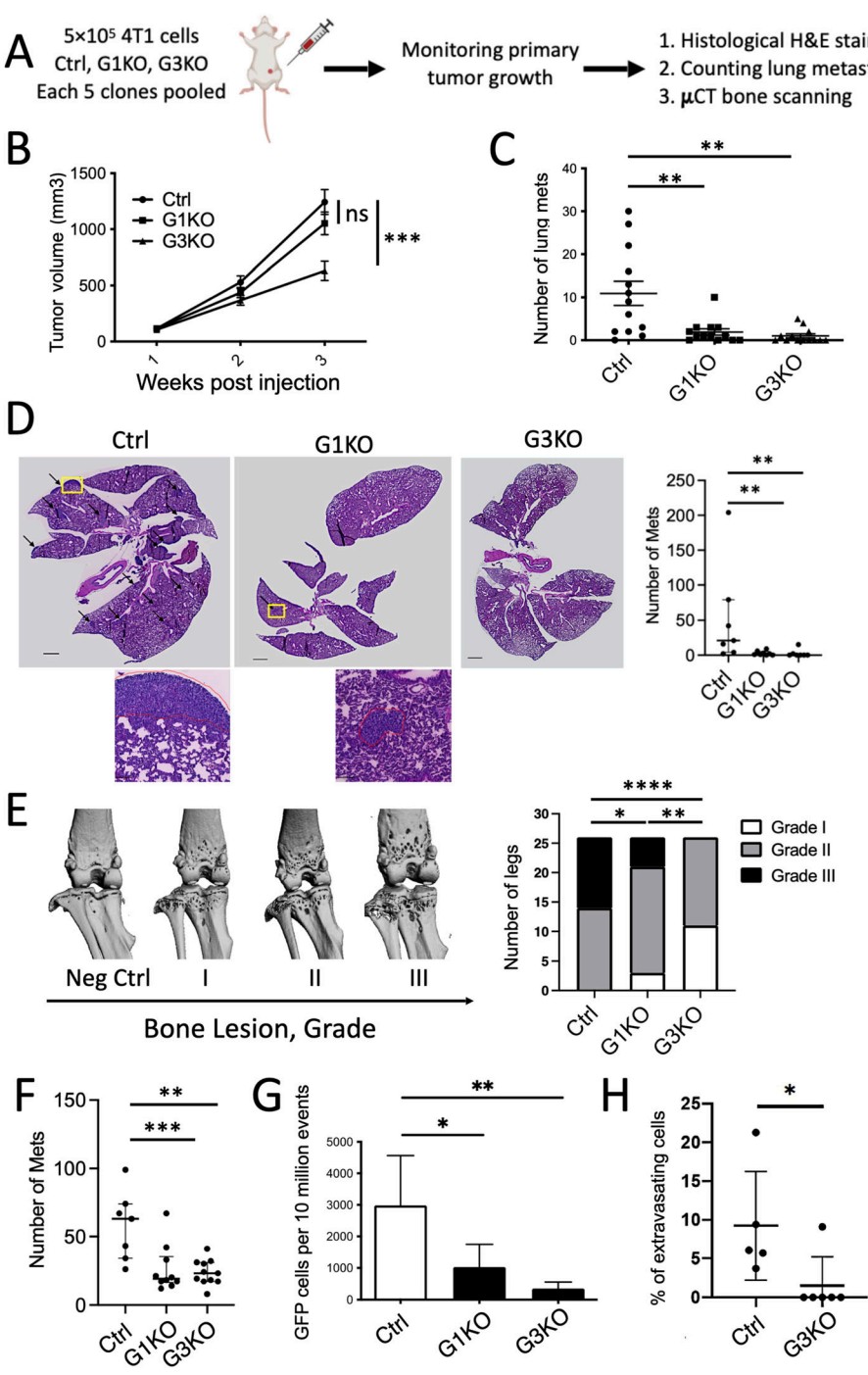

**Figure 1. SWAP-70 promotes metastasis in mice.**
**(A)** Scheme showing the workflow of the mouse breast cancer metastasis experiments. **(B)** Growth curve of primary tumors in BALB/c mice injected with 4T1 Ctrl, G1KO, or G3KO cells, n = 13 in each group. Each point is the mean tumor volume at each time point ± SEM. Statistical differences were tested by an unpaired two-tailed $t$ test (***$P <$ 0.001). **(C)** Quantification of metastatic nodules per lung of different cell lines in BALB/c mice, n = 13 in each group. Statistical differences were tested by an unpaired two-tailed $t$ test (**$P <$ 0.01). **(D)** Representative histological images of lung sections; the lung metastases are marked with arrows. High-magnification insets of the areas indicated by yellow boxes are provided for ctrl and G1KO, there was no tumor in G3KO. Quantification of lung metastasis from the histological sections. Ctrl, n = 7; G1KO, n = 8; and G3KO, n = 7. Statistical differences were tested by an unpaired two-tailed $t$ test (**$P <$ 0.01). **(E)** Representative images of the bones as scanned by μCT, and associated bone lesion levels, grades I to III. Bone specimens were scored blindly based on the lesion level; the statistical analysis was performed by the Mann–Whitney test, n = 26 in each group (*$P <$ 0.05, **$P <$ 0.01, ***$P <$ 0.001, and ****$P <$ 0.0001). **(F)** Quantification of bone metastasis determined by histology. Ctrl, n = 10; G1KO, n = 10; and G3KO, n = 11. Statistical differences were tested by an unpaired two-tailed $t$ test (mean; **$P <$ 0.01 and ***$P <$ 0.001). **(G)** Quantification of the number of GFP-positive cells per 10 million events present in the lungs 16 h after tail vein injection. Statistical analysis was done by an unpaired two-tailed $t$ test, n = 6 in each group (mean ± SD; *$P <$ 0.05 and **$P <$ 0.01). **(H)** Percentage of extravasating cells is shown for control and G3KO cells after CMFDA labeling of the cells, tail vein injection, and quantification by imaging; n = 5 for Ctrl; n = 6 for G3KO; unpaired $t$ test (*$P <$ 0.05).

2.2-fold difference between Ctrl and G3KO with no significant difference between G1KO and G3KO (Figs 1F and S1B). Despite the difference between G1KO and G3KO in the rate of primary tumor growth, both produced fewer distant metastasis to either lung or bone. The one moderately distinct phenotype is osteolysis, a secondary effect of metastasis that was more enhanced in G1KO cells. The very similarly reduced capacity to form metastases suggested that both KO cell pools are less capable of disseminating from the primary tumor, of traveling to a distant tissue, and/or of homing into bone.

To specifically test whether the KO-derived tumor cells failed to home into the lungs, we injected GFP-labeled Ctrl, G1KO, or G3KO cell pools into the tail vein of BALB/c mice and monitored the presence of tumor cells in the lung after 16 h by flow cytometry. This tumor cell burden is presented as the number of GFP-positive cells per 10 million events. The Ctrl group showed a threefold and an eightfold higher number of GFP-positive cells compared with the G1KO and G3KO groups, respectively (Figs 1G and S1C), indicating that SWAP-70 is crucial for blood vessel–borne tumor cells to

accumulate in the lung. To assess whether SWAP-70–deficient cells extravasate as efficiently as controls, we used CMFDA-labeled tumor cells in the same setup except that after 18 h lungs were collected, sectioned, and stained with anti-CD31 to label blood vessels. Confocal microscopic images, i.e. z-stacks, were collected to examine the spatial relationship between the labeled tumor cells and the vessels. Cells that are either outside vessels or are reaching with protrusions through vessel walls were counted as extravasated/extravasating. The results show a greatly impaired ability of SWAP-70–deficient tumor cells to extravasate; for except one mouse with only two extravasating/extravasated cells detected, there were no extravasating mutant cells (Figs 1H and S1D).

To understand a possible impact of the genetic background of the mouse strain and whether SWAP-70 had the same impact in a different tumor model system, we generated SWAP-70–deficient Lewis lung carcinoma (LLC) cells, a well-established tumor model in the C57BL/6 mouse strain, using the same CRISPR/Cas9 method as described for the 4T1 cells. The absence of SWAP-70 was confirmed by immunoblotting (S1E). Cell lines derived from single G1KO, G3KO clones, or mock-treated Ctrl cells were injected subcutaneously into the back of the mice. Tumor growth and metastasis were monitored, and mice were euthanized under the conditions explained for the 4T1 model. The number of distant lung metastasis was determined by H&E staining of paraffin-embedded lung sections. As with the 4T1 orthotopic model, there was significantly less metastasis generated by the two SWAP-70–deficient clones, with G1KO showing 2.6-fold less metastasis and G3KO presenting 1.9-fold reduced metastasis compared with Ctrl. There was no significant difference between G1KO- and G3KO-injected mice (Fig S1F).

These data collectively suggest that SWAP-70 strongly supports metastatic tumor formation from primary tumors.

## SWAP-70 controls tumor cell, migration, invasion, adhesion, and morphogenesis

It is well known that growth factors actively participate in tumorigenesis by activating complex signaling pathways, and regulating cancer cell proliferation, division, migration, and invasion (Sever & Brugge, 2015). TGFβ and EGF are two of the primary growth factors that regulate invasion and metastasis of breast cancer tumors and promote EMT (Thiery, 2002; Kalluri & Weinberg, 2009). Both growth factors are commonly used as chemoattractants in cancer cell migration assays. The combination of TGFβ and EGF is a potent inducer of cytoskeletal reorganization and tumor cell migration (Uttamsingh et al, 2008; Buonato et al, 2015). To test features of tumor cells relevant for metastasis, several functional assays were performed on Ctrl and KO cell pools before and after TGFβ/EGF stimulation.

Before such specific assays, we assessed a potential role of SWAP-70 in cell viability with or without TGFβ/EGF stimulation; we costained the cells with 4′,6-diamidino-2-phenylindole (DAPI) and Annexin V reflecting early and late apoptotic cells, respectively. No significant difference was observed between Ctrl and KO cells regardless of TGFβ/EGF treatment (Fig S2A). Cell proliferation was also not different between Ctrl and KO lines (all lines doubled in cell number app. 2.5 times within 2 d).

Furthermore, a putative effect of SWAP-70 deficiency on EMT was investigated by several approaches. The expression of two transcription factors that drive EMT early in response to TGFβ/EGF, Snail and Twist (Brabletz et al, 2021), was assessed by qRT–PCR (Fig S2B). Neither at 12 or 24 h after TGFβ/EGF treatment were levels of Snail and Twist mRNA significantly different between Ctrl and the two KO cell pools. Next, we examined several reported mouse breast cancer surface markers indicative of EMT (Pastushenko & Blanpain, 2019) by flow cytometry. Intermediate EMT marker CD106 and full EMT marker CD51 and CD61 proteins were up-regulated on the surface of all of the cell pools after TGFβ/EGF treatment for 3 d (Fig S2C) and were the same for Ctrl and KO cell pools before stimulation. Although the level of CD61 expression was slightly lower in unstimulated G1KO versus control, after stimulation no significant difference was seen. All other markers were the same between Ctrl, G1KO, and G3KO, indicative of similar responsiveness to TGFβ/EGF of the three cell pools with respect to EMT. This was also observed when testing proliferation, apoptosis, and other activities as specified below, indicating unaltered general functionality of the TGFβ/EGF pathways. All cell lines also showed similar levels of the epithelial cell marker EpCAM regardless of TGFβ/EGF treatment (Fig S2C). In addition, we used RNA sequencing to profile the transcriptome of Ctrl and KO cell pools. We profiled unstimulated cells and cells that were stimulated with TGFβ/EGF for 72 h (Fig S3A). Gene set enrichment analysis identified EMT pathway genes that are very similarly up-regulated under stimulated conditions (Fig S3B), starting from lower gene set enrichment in unstimulated KO cells (Fig S3C), which did not translate into the base status of the above-described EMT proteins. Principal component analysis showed that treatment with TGFβ/EGF induced similar changes to the transcriptome for each of the genotypes (Fig S3D). Taken together, SWAP-70 does affect neither cell viability nor up-regulation of EMT. Therefore, we focused on cell properties directly related to the F-actin modulatory functions of SWAP-70.

To investigate the effects of SWAP-70 on anchorage-independent cell growth of 4T1 cells, a soft agar colony formation assay was performed. Ctrl and KO cell pools were cultured in complete DMEM supplemented without or with TGFβ/EGF present throughout the experiment (Figs 2A and S2D). Without growth factors, the three cell lines showed no differences in colony formation. Upon TGFβ/EGF treatment, Ctrl cells showed threefold and 1.5-fold more colony formation compared with the G1KO and G3KO cells, respectively.

Migration is an initial and crucial step for cancer cells to invade tissues. We used the Oris Cell Migration tissue culture wound-healing assay to assess two-dimensional (2D) migration of each 4T1 cell pool. Without TGFβ/EGF treatment after 15 h of live-cell imaging, all cell lines displayed similar migration into the cell-free area (Fig 2B). Upon TGFβ/EGF treatment, Ctrl cell migration into the open area increased ~twofold, whereas migration of both KO cell pools was not enhanced.

We also measured the cells' three-dimensional (3D) migratory response using the Boyden chamber Transwell assay with TGFβ and EGF as the chemoattractants (Figs 2C and S4A). Regardless of TGFβ/EGF treatment, the number of Ctrl cells transmigrating through the membrane was up to threefold higher than the number of migrating KO cells. TGFβ and EGF strongly enhanced the mobility of all cell pools, and the two KO cell pools did not differ from each other

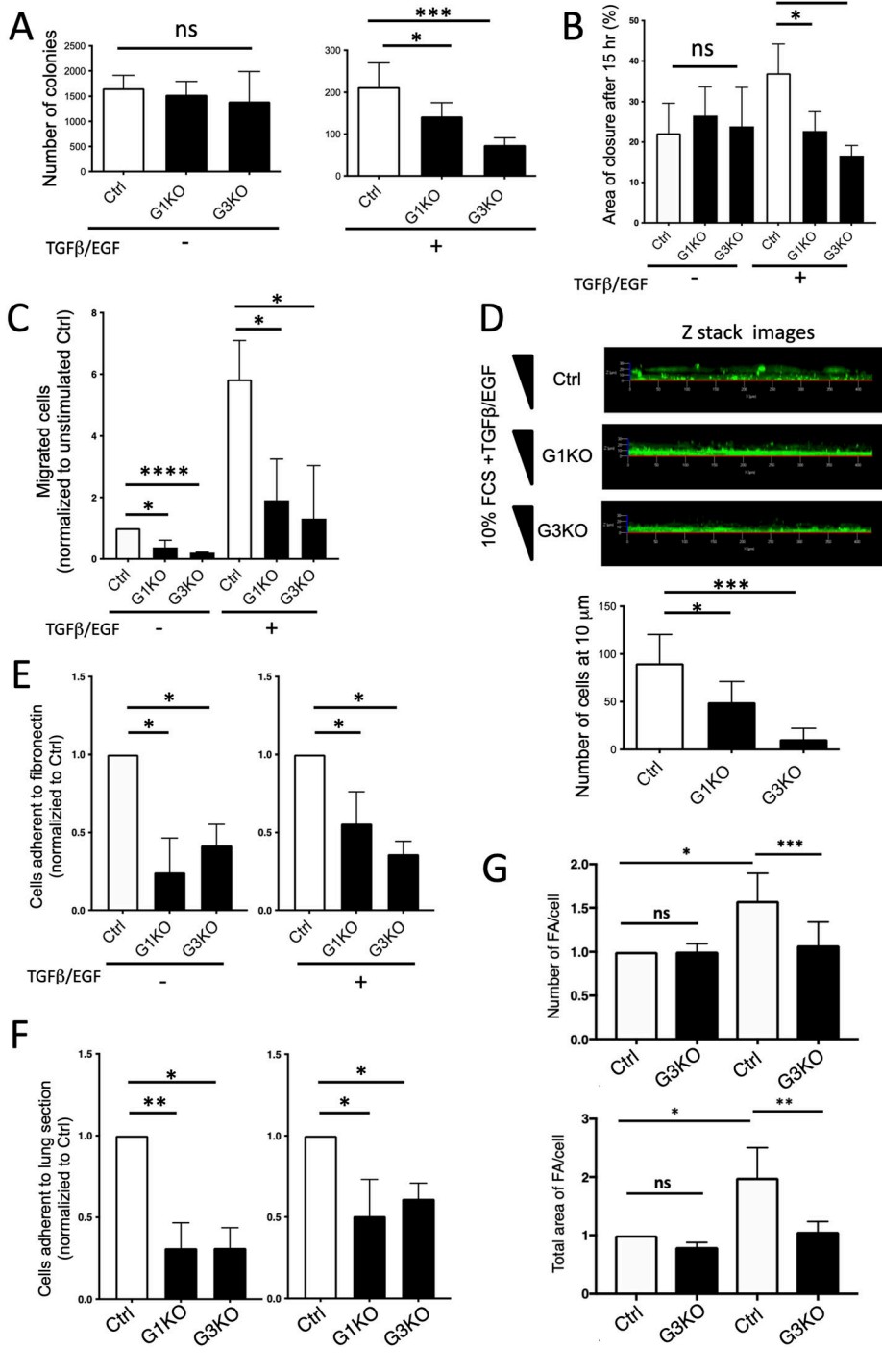

**Figure 2. SWAP-70–dependent cell features relevant for metastasis.**

**(A)** 4T1 cell lines were subjected to the soft agar colony formation assay, and the colonies were stained with crystal violet at day 15. Quantification of colony numbers per well is presented in the bar graph. Data are represented as means ± SD from three independent experiments. Statistical differences were tested with an unpaired two-tailed $t$ test (*$P < 0.05$ and ***$P < 0.001$). **(B)** 2D cell migration assayed using Oris Cell Migration Assay in fibronectin-coated 96-well plates. The summary bar graph illustrates the percentage of wound closure after 15 h. Data are presented as means ± SD from four independent experiments, and statistical differences were tested with an unpaired two-tailed $t$ test (*$P < 0.05$ and **$P < 0.01$). **(C)** 3D cell migration in Boyden chambers. The transmigrated cells were fixed and then stained by DAPI 24 h after seeding into the well. Complete DMEM with or without TGF$\beta$ and EGF was used as a chemoattractant. A summary bar graph is shown illustrating the fold change of the migrated cells normalized to the non-treated Ctrl. Data are presented as means ± SD from three independent experiments, and statistical differences were tested by a ratio-paired two-tailed $t$ test (*$P < 0.05$ and ****$P < 0.01$). **(D)** Invasion assay performed using a Matrigel-coated Transwell chamber. The transmigrated and invaded cells were stained with CellTracker Green CMFDA dye at 72 h and imaged by confocal microscopy. EGF and TGF$\beta$ in complete DMEM were used as chemoattractants, and the concentration gradient is indicated by a triangle. Representative Z-stack images are shown with cells that passed the membrane and invaded into the Matrigel. The number of cells reaching 10-$\mu$m distance from the membrane was counted. Data are presented as means ± SD from three independent experiments, and statistical differences were tested by an unpaired two-tailed $t$ test (*$P < 0.05$ and ***$P < 0.001$). **(E)** Adhesion of 4T1 cell lines to fibronectin-coated wells. The number of GFP+ cells adhered to the surface was ascertained. A summary bar graph illustrates the fold change of the cells normalized to the EGF/TGF$\beta$-untreated Ctrl or EGF/TGF$\beta$-treated Ctrl, resp., as means ± SD from at least three independent experiments, and statistical differences were tested by a ratio-paired two-tailed $t$ test (*$P < 0.05$). **(F)** Adhesion of 4T1 cell lines to mouse lung sections. A summary bar graph illustrates the fold change of adherent cells normalized to the EGF/TGF$\beta$-untreated Ctrl or EGF/TGF$\beta$-treated Ctrl, resp., with SD shown as means from at least three independent experiments, and statistical differences were tested by a ratio-paired two-tailed $t$ test (*$P < 0.05$ and **$P < 0.01$). **(G)** Formation of focal adhesions (FA) by control and SWAP-70–deficient 4T1 cells, 20 cells per sample were assessed by staining for vinculin, and the number of focal adhesions and the area covered by them were quantified. Statistical differences were tested by a ratio-paired two-tailed $t$ test (n = 4; *$P < 0.05$, **$P < 0.01$, and ***$P < 0.005$).

significantly. In addition, Ctrl cells demonstrated a significantly higher invasive capacity in a Matrigel-coated Transwell system (Fig 2D). 3D reconstructed images demonstrated how cells not only passed through the Transwell membrane but also invaded the Matrigel and migrated upward. Cells that reached a distance of 10 $\mu$m from the membrane were counted, and G1KO and G3KO cell pools displayed twofold and ninefold lower Matrigel invasion compared with Ctrl (Fig 2D). Here, G3KO invaded even less efficiently into the Matrigel compared with the G1KO.

Next, we assessed adhesion of the cells to fibronectin and to fixed mouse lung sections. We ensured that GFP labeling of the cells did not affect adhesion. Regardless of stimulation, the stained Ctrl-

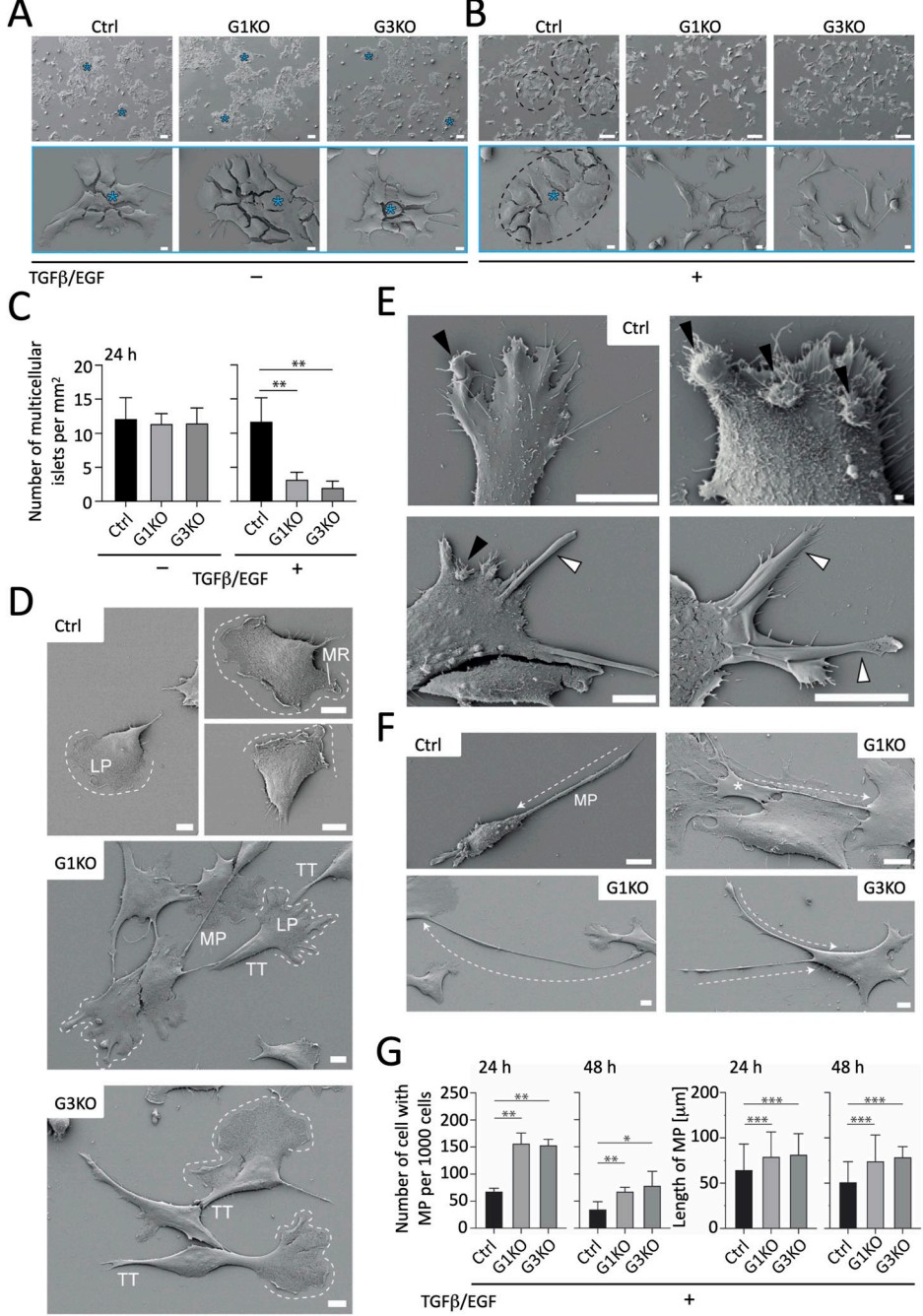

**Figure 3. SWAP-70–dependent ultrastructure of 4T1 cells.**

**(A, B, C, D, E, F, G)** Morphological characteristics of the indicated cells were observed at high resolution by SEM without or with TGFβ/EGF treatment. **(A, B, C, D, E, F, G)** Cells were incubated for 24 (A, B, C, D, E, F, G) or 48 (G) h after seeding on fibronectin-coated glass coverslips. **(A, B, C)** As observed at low-power magnification, all cell lines were developed into multicellular islets ((A, B), blue asterisk, dashed line) containing at least five cells in the absence of treatment, whereas only cells in the control group (Ctrl) retained this characteristic after TGFβ/EGF treatment. **(C)** Statistical analysis was performed using the Mann–Whitney test, n = 4–6 independent micrographs [area of 2.67 mm$^2$ each] for each group, **$P < 0.01$). At high-power magnification, individual cells show distinct migratory morphology between the Ctrl and KO cell pools. **(D)** Ctrl cells have lamellipodia (LPs, dashed line) with membrane ruffles (MR) on their leading edge that has a convex shape, whereas SWAP-70–deficient cells have smooth LPs without ruffles that are extended relative to the cell body and often indented. **(E)** Although the latter phenotypes can be rarely observed in Ctrl cells (about 1:100), they harbor circular membrane ruffles and/or folded protruding lamella structures, black and white arrowheads, respectively; four examples of Ctrl are displayed). At the rear, Ctrl cells contain very short retraction processes, whereas they appear as a long trailing tail (TT) in KO cells. **(D, F)** These processes can be very elongated and look like magnupodium-like structures (MP), dashed arrow). **(F)** MPs may harbor an LP at their extremity (white asterisk) and make contact with other cells. **(G)** In TGFβ/EGF-treated cells, MPs are predominant and longer in the KO cell pools, as observed after 24 and 48 h of culture. **(G)** Statistical analysis was performed using the Mann–Whitney test; cell number: n > 2100 cells; length: n > 25 cells, *$P < 0.05$, **$P < 0.01$, and ***$P < 0.001$). Scale bars, 100 μm ((A, B) top panels); 10 μm ((A, B) bottom panels, (D, E) except top right panel, and (F)); 1 μm ((E), top right panel).

GFP cells did not behave significantly differently from non-GFP Ctrl cells (Fig S4B). With or without TGFβ/EGF treatment, Ctrl cells displayed higher adhesion to fibronectin than KO cells (threefold higher than unstimulated cells and twofold higher than stimulated cells) (Figs 2E and S4C). Frozen mouse lung sections provide a semi-physiological substrate allowing the 4T1 cell lines to interact with other cell types and adhesion molecules. The results reflected those obtained after adhesion to fibronectin, in that unstimulated Ctrl cells adhered threefold more than unstimulated KO cells, and after TGFβ/EGF stimulation, twofold more Ctrl cells adhered than stimulated KO cells (Figs 2F and

S4D). We also analyzed formation of focal adhesions using vinculin as a marker as they link the actin cytoskeleton to sites of integrin-mediated adhesion (Kanchanawong & Calderwood, 2023). Upon stimulation by TGFβ/EGF, SWAP-70–deficient tumor cells failed to increase the number of focal adhesions and the area that they cover (Figs 2G and S4E), consistent with the adhesion deficiencies of the mutant cells.

Collectively, these findings of mouse and cell culture experiments show that SWAP-70 supports metastasis of 4T1 cells in vivo correlating with the protein's promotion of colony formation, migration, invasion, and adhesion.

The formation of cell–cell contacts and cell shape affects the above-described cell features and depends on F-actin structures. Thus, we next analyzed morphological and ultrastructural features of the cell lines with or without TGFβ and EGF stimulation via scanning electron microscopy (SEM). Without TGFβ/EGF, the data revealed no significant differences between Ctrl and KO cells regarding the overall cell shape and frequent growth into multicellular colony-like islets (Fig 3A). However, cell distribution changed significantly after 3 d of TGFβ/EGF treatment. Although the clusters of Ctrl cells remained detectable, both KO cell pools grew as much more dispersed single cells (Fig 3B and C). At the cellular level, their shapes were also distinct, although all cells with a migratory profile showed a morphology typical of a mesenchymal migration mode (Shafqat-Abbasi et al, 2016). Most individual Ctrl cells harbored a large lamellipodium with membrane ruffles on a convex leading edge, whereas in both groups of KO cells, the smooth and extended lamellipodium had numerous indentations (Fig 3D). It should be noted that Ctrl cells showing indented lamellipodia contain circular membrane ruffles and folded protrusive lamella structures at their edge that appear as smooth rod-like structures (Fig 3E). These dynamic structures that are absent in KO cells are implicated in active cell motility and/or internalization processes and are induced in response to growth factors (Hoon et al, 2012). The presence of these structures is consistent with the migration of Ctrl cells (see above) and potential role of SWAP-70 in macropinosomes and phagosomes. Quantifying membrane ruffles by phalloidin staining of Ctrl and KO cells confirmed the reduced presence of ruffles, app. threefold reduced, in KO cells (Fig S5A). The backward structure that is equivalent to the trailing tail was also shorter in Ctrl cells than in KO cells, suggesting a failure (or delay) of its retraction in SWAP-70–deficient cells (Fig 3E). Such feature is also reflected in the appearance of long magnupodium-like protrusions that are observed more frequently in KO cells (Fig 3F and G). These long structures may or may not have a lamellipodium-like structure at their extremity and contact other cells, thus interfering with their migration (Fig 3E). Overall, the morphological differences between Ctrl cells and KO cells upon growth factor stimulation may suggest that cells lacking SWAP-70 are less dynamic and possibly stiffer, which could negatively influence cell migration and adhesion, both essential features for the metastatic process.

The reduced ability of SWAP-70–deficient tumor cells to form metastatic tumors may, in part, be caused by impaired capacity to degrade extracellular matrix. Thus, we measured the activity of secreted matrix metalloproteases by zymography and degradation of a gelatin matrix by the tumor cells. Neither protease activity (Fig S5B) nor matrix degradation (Fig S5C) was reduced in the absence of SWAP-70, regardless of whether the cells were left unstimulated or stimulated by TGFβ/EGF.

## SWAP-70 modulates biophysical features of tumor cells

Recent studies showed how the mechanical properties of cells contribute to the spread of cancer, with malignant cells typically being softer than non-malignant less aggressive cells (Ghosh & Dawson, 2018). Because cancer cells need to transmigrate into and out of vessels to invade tissues, the cell body needs to be sufficiently flexible to squeeze through confined regions or tight

junctions. F-actin rearrangements are required to shape cell rigidity by regulating the stiffness and tension of the actin cortex. Therefore, we hypothesized that SWAP-70 might regulate cellular stiffness via the interaction with F-actin. We applied atomic force microscopy (AFM) to measure cellular cortical stiffness and tension at the single-cell level (Hosseini et al, 2020a; Hosseini et al, 2022). The AFM results revealed significantly higher cortical stiffness (Fig 4A), tension (Fig 4B), and phase shift (Fig S6A) of both KO cell pools compared with Ctrl after treatment with TGFβ/EGF, but no changes in the cell volume (Fig S6B). No or very mild significant differences were observed without TGFβ/EGF treatment. G3KO were more severely affected than G1KO displaying higher cortical stiffness and tension compared with G1KO. This higher cell rigidity may explain why G3KO were less invasive in the 3D invasion assay (Fig 2D).

To further understand how SWAP-70 modulates the mechanical properties of cells after stimulation with TGFβ and EGF, untreated and treated cells were transfected with plasmids expressing either an actin-mCherry or myosin light chain-9-mApple (Fig 4C and D). We used confocal microscopy, imaged the equatorial cross-section of the suspended cells, and calculated the ratio of cortical versus cytoplasmic fluorescence intensity reflecting actin or myosin distribution. The results showed that the cortex-to-cytoplasm ratio of actin and myosin was significantly higher in both KO cell pools after the EGF/TGFβ treatment (Fig 4D). These data suggest that more F-actin and myosin were redistributed into the cortex of the KO cells upon TGFβ and EGF induction, which correlates with the higher cortical tension and stiffness measured by AFM (Hosseini et al, 2020b).

## Binding to F-actin is required for SWAP-70–mediated cell migration, adhesion, and cell mechanics

Because of its role as an F-actin binding and bundling protein, it is critical to study whether the interaction of SWAP-70 with F-actin is required for the phenotypes of SWAP-70–deficient cells described herein. We performed complementation experiments by introducing SWAP-70 (S70) or SWAP-70 F-actin binding mutant (ABM)–expressing plasmids in both G1KO and G3KO cell lines. The SWAP-70 ABM construct features a deletion (Δ527–585) of the F-actin binding site, which prevents SWAP-70 from interacting with F-actin (Ocana-Morgner et al, 2013). SWAP-70 expression in these cell lines was confirmed by immunoblotting (Fig S6C). In most of the above in vitro assays, the phenotypic differences between the Ctrl and KO cells surfaced upon TGFβ/EGF treatment. Thus, the complementation experiments were performed in the presence of TGFβ/EGF. In Transwell migration experiments, cells expressing full-length SWAP-70 protein, that is, G1KO-S70 and G3KO-S70, showed comparable migratory phenotypes to the Ctrl cells, and thus, phenotypic rescue was achieved (Fig 5A). However, the KO cell pools expressing the SWAP-70 ABM, G1KO-ABM and G3KO-ABM, showed no rescue of the SWAP-70 KO phenotype (Fig 5A). This suggests that the interaction between SWAP-70 and F-actin is crucial for this type of cell motility. Furthermore, the G1KO-S70 and G3KO-S70 cells showed very similar adhesion to fibronectin compared with Ctrl cells and thus phenotypic rescue. In contrast, G1KO-ABM and G3KO-ABM cell lines displayed poor cell adhesion, similar to KO cells (Fig 5B). The increase seen for KO cells in cortical stiffness (Fig 5C) and cortical

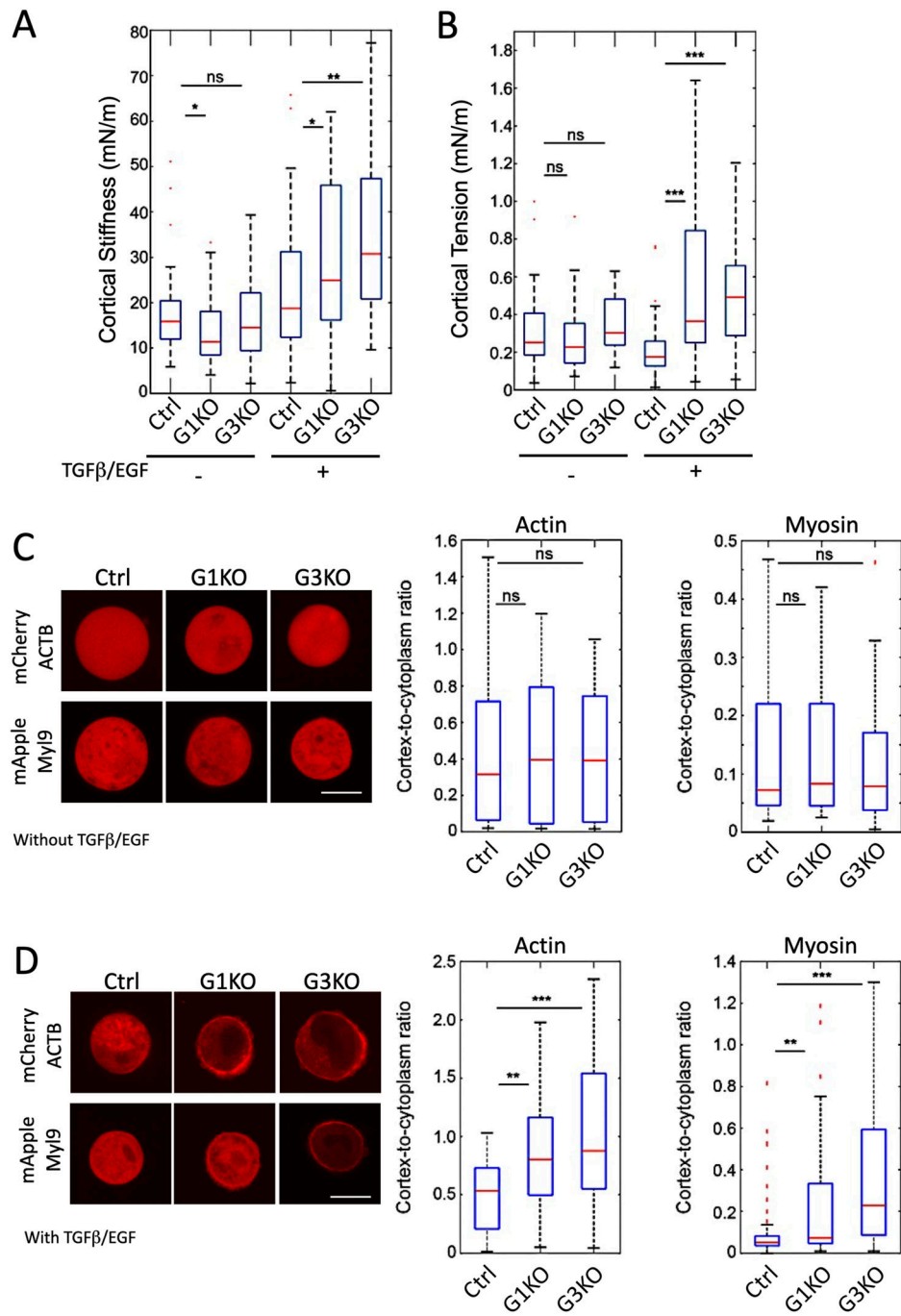

**Figure 4. SWAP-70 determines the mechanical properties of 4T1 cells.**
**(A, B)** Cortical stiffness and (B) cortical tension of the suspended cells were measured by AFM, either without or with TGFβ/EGF treatment for 48 h; untreated Ctrl, n = 40; G1KO, n = 32; and G3KO, n = 24; treated Ctrl, n = 30; G1KO, n = 30; and G3KO, n = 29. Measurements are from two independent experiments, and statistical differences were calculated by a two-tailed Mann–Whitney *U* test (ns, not significant, *P < 0.05, **P < 0.01, and ***P < 0.001). **(C)** Representative confocal images of TGFβ/EGF-untreated suspended cells expressing mCherry-ACTB or mApple-Myl9. A summary box plot of the ratio of cortical versus cytoplasmic actin (left) or myosin (right) of TGFβ- and EGF-untreated cells. The number of cells measured for actin analysis: Ctrl, n = 44; G1KO, n = 43; and G3KO, n = 48; for myosin: Ctrl, n = 48; G1KO, n = 48; and G3KO, n = 48. Measurements are from two independent experiments, and statistical differences were calculated by a two-tailed Mann–Whitney *U* test (ns, not significant). **(D)** Representative confocal images of TGFb- and EGF-treated suspended cells expressing mCherry-ACTB or mApple-Myl9. A summary box plot of the ratio of cortical versus cytoplasmic actin (left) or myosin (right) of TGFβ/EGF-untreated cells. The number of cells measured for actin analysis: Ctrl, n = 46; G1KO, n = 43; and G3KO, n = 43; for myosin: Ctrl, n = 60; G1KO, n = 62; and G3KO, n = 60. Measurements are from two independent experiments, and statistical differences were done by a two-tailed Mann–Whitney *U* test (**P < 0.01 and ***P < 0.001). Scale bars: 10 μm.

tension (Fig 5D) was abolished in the G1KO-S70 and G3KO-S70 cells, proving that in the presence of SWAP-70, stimulated 4T1 cells are softer.

Similarly, we calculated the ratio of cortical versus cytoplasmic fluorescence intensity of actin or myosin in the complemented cell pools. As expected, G1KO-S70 and G3KO-S70 cells showed redistributed actin and myosin from the cortex to the cytoplasm, with a similar ratio as in the Ctrl cells, and thus rescue. However, in cells expressing SWAP-70 ABM, we saw an increase in the ratio of cortical versus cytoplasmic actin and myosin even beyond that of KO cells, suggesting that the SWAP-70 actin binding mutant cannot rescue and may even have a dominant negative effect (Fig 5E and F).

Besides AFM, we also employed real-time deformability cytometry (RT-DC) to further characterize cell mechanical properties in a larger population, as this technology allows measuring certain parameters of tens of thousands of cells. RT-DC is a high-throughput assay, measuring cell deformation and size of

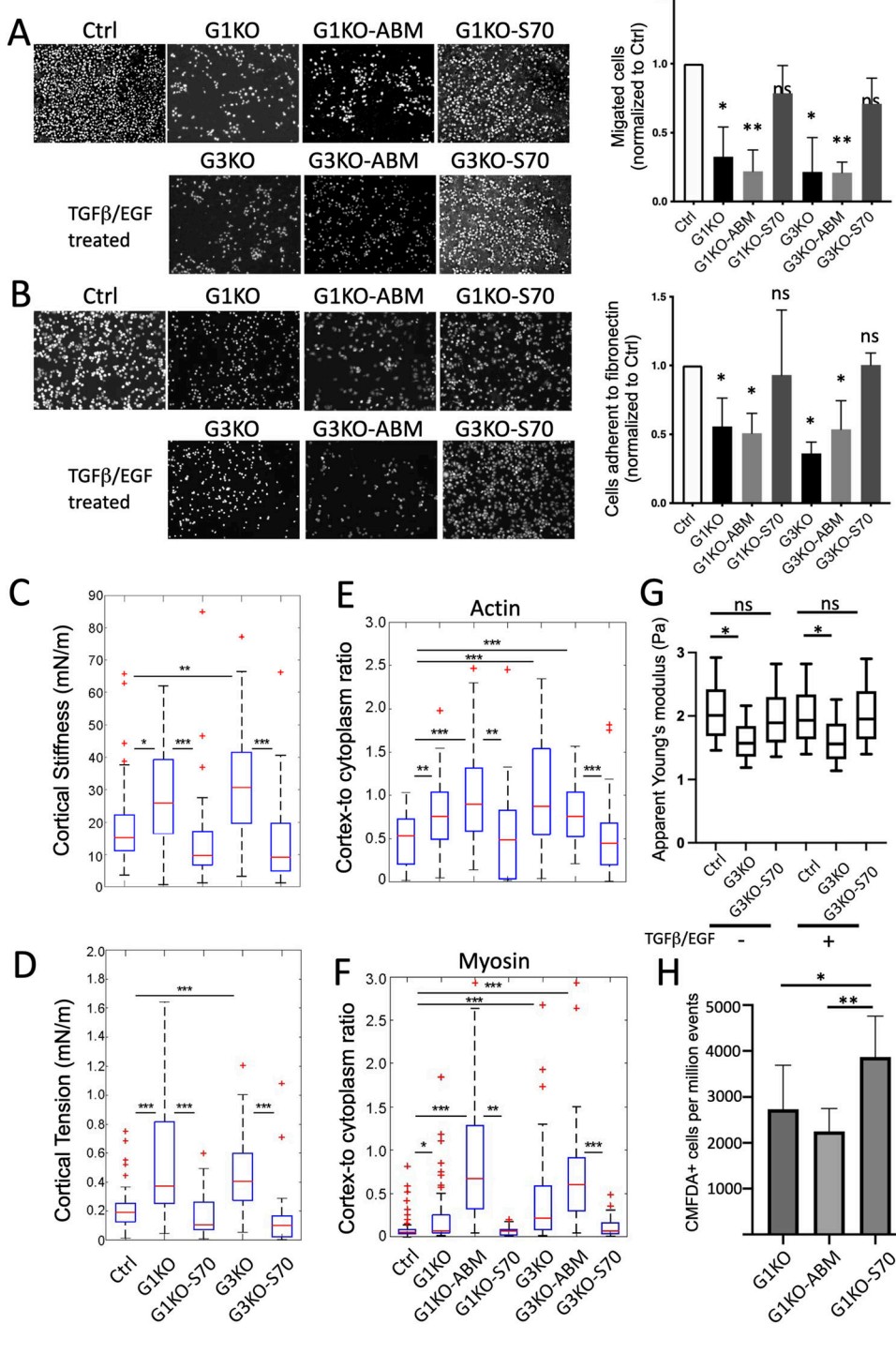

**Figure 5. SWAP-70 regulates cell migration, adhesion, and mechanics by interacting with F-actin.**

**(A)** 3D migration analyzed in a Boyden chamber. The transmigrated cells were fixed and then stained by DAPI at 24 h for visualization. Complete DMEM with TGFβ/EGF was used as a chemoattractant. A summary bar graph illustrating the fold change of the migrated cells normalized to the Ctrl. Data are presented as means ± SD from at least three independent experiments, and statistical differences were tested by a ratio-paired two-tailed *t* test (ns, not significant, *P < 0.05, and **P < 0.01). **(B)** Adhesion of 4T1 cell lines to a fibronectin-coated surface. A summary bar graph illustrating the fold change of the cells normalized to the Ctrl as means ± SD from at least three independent experiments, and statistical differences were tested by a ratio-paired two-tailed *t* test (ns, not significant and *P < 0.05). **(C, D)** Cortical stiffness and (D) cortical tension of the suspended cells were measured by AFM after TGFβ/EGF treatment for 48 h. The number of cells measured in the treated Ctrl, n = 55; G1, n = 27; G1-S70, n = 32; G3, n = 27; and G3-S70, n = 34. Measurements represent at least two independent experiments, and statistical differences were done by a two-tailed Mann–Whitney *U* test (ns, not significant, *P < 0.05, **P < 0.01, and ***P < 0.001). **(E, F)** Summary box plot of the ratio of cortical versus cytoplasmic actin (E) and myosin (F) of TGFβ- and EGF-treated cells. The number of cells analyzed: actin: Ctrl, n = 46; G1, n = 36; G1-ABM, n = 42; G1-S70, n = 63; G3, n = 29; G3-ABM, n = 31; G3-S70, n = 62; myosin: Ctrl, n = 57; G1, n = 58; G1-ABM, n = 45; G1-S70, n = 38; G3, n = 56; G3-ABM, n = 47; G3-S70, n = 43. Measurements represent at least two independent experiments, and statistical differences were done by a two-tailed Mann–Whitney *U* test (*P < 0.05, **P < 0.01, and ***P < 0.001). **(G)** Real-time deformability cytometry analysis of the 4T1 cell lines. Apparent Young's moduli are presented as box-and-whisker plots (box presents 25th and 75th quartiles and median, and whiskers indicate 10th and 90th percentiles). Five independent experiments were performed, and statistical differences were done using Tukey's test. (ns, not significant and *P < 0.05). **(H)** Accumulation in the lungs of G1KO cells or G1KO cells re-expressing either full-length SWAP-70 (G1KO-S70), or the actin binding deficient mutant of SWAP-70 (G1KO-ABM). Accumulation was compared by quantifying the number of CMFDA-positive 4T1 cells per 1 million events present in the lungs 18 h after tail vein injection. Statistical analysis was done by an unpaired two-tailed *t* test, n = at least seven mice in each group (mean ± SD; *P < 0.05 and **P < 0.01).

suspended cells (>100 cells/s, thus msec per cell) under hydrodynamic shear stress and pressure in a microfluidic chip (Otto et al, 2015). Using numerical and analytical models, the apparent Young's modulus is calculated (Mokbel et al, 2017), a measure of elasticity under tension or compression. The results showed lower Pascal values for G3KO cells, suggesting that G3KO cells were more deformable; that is, under these conditions, they were not as stiff compared with the Ctrl cells, regardless of EGF/TGFβ treatment. The complemented G3KO-S70 showed a reversed phenotype comparable to the Ctrl cells (Fig 5G). This demonstrates that SWAP-70 affects cell deformability and shape under high fluidic velocity and mechanical stress.

To assess whether actin binding of SWAP-70 is also required for efficient homing into the lung in vivo, we asked whether re-expressing of SWAP-70 in SWAP-70–deficient 4T1 cells would restore their ability to home into the lung and whether the expression of a SWAP-70 mutant not able to bind F-actin (G1KO-ABM) would fail to do so (Figs 5H and S6D). CMFDA-labeled cells present in the lungs of BALB/c mice 18 h after tail vein injection were quantified by FACS. There was a mild restoration of lung homing by the expression of full-length SWAP-70, but cells that re-expressed the actin binding deficient SWAP-70 not only failed to rescue lung homing but were significantly fewer and may have shown even a mild dominant negative phenotype.

In summary, these data show that the SWAP-70 F-actin binding domain and thus the interaction of SWAP-70 with F-actin are critical for cell migration, adhesion, cell mechanics, and homing of the tumor cells into the lung.

# Discussion

Aberrant rearrangements of the F-actin cytoskeleton are a prominent feature of many tumor cells and are particularly important for metastasis. Thus, deciphering key features of metastasis requires understanding the roles of F-actin and its associated proteins. A number of F-actin modulatory proteins have been investigated in this context and shown to often be hijacked during tumor cell evolution to the benefit of these malignant cells (Olson & Sahai, 2009; Gross, 2013; Izdebska et al, 2020). We undertook an investigation of the unique protein SWAP-70's role in metastasis both in in vivo and in cultured cells. SWAP-70 binds and bundles F-actin but differs from all other known F-actin bundling proteins in domain composition. Other F-actin bundling proteins have been shown to be involved in tumor cell metastasis; notably, fascin has been reported to have an important role in promoting tumor cell invasion (Adams, 2004; Machesky & Li, 2010). However, fascin differs in several respects from SWAP-70. Fascin features two F-actin binding sites, whereas SWAP-70 carries one, requiring SWAP-70 to multimerize for bundling F-actin (Chacon-Martinez et al, 2013). The domain structure of the two proteins is very different, and unlike fascin, SWAP-70 is expressed only in vertebrates. Thus, a hitherto undescribed metastasis-promoting protein and the associated metastasis-related pathways are described in this communication.

SWAP-70–deficient breast cancer cells are impaired in migration as several assays showed. Our biophysical analysis of cytoskeletal-associated features of these cells suggests an explanation: without SWAP-70, the cells are stiffer and thus less able to acquire a motile phenotype, to efficiently heal cell monolayer wounds, to quickly squeeze through pores such as in Transwell setups, and to invade extracellular matrix such as Matrigel. The rare differences seen between the two KO cell pools, G1KO and G3KO, can largely be explained by the higher level of stiffness of the G3KO cells, which invade Matrigel somewhat less efficiently. These phenotypes are unlikely to be because of off-target effects of the CRISPR/Cas9 as restoration of SWAP-70 expression in mutant cells restores function.

Because changes in cytoskeletal dynamics in tumor cells can correlate with the EMT, which can be triggered by TGFβ/EGF used in the cellular assays in our report, we analyzed the requirement for SWAP-70 for the expression of EMT markers by several approaches. No significant differences were observed either early (mRNA) or later (surface markers) in these markers. We conclude that SWAP-70 is not prominently involved in the control of the EMT.

To migrate, tumor cells need to build up force to overcome opposing processes such as adhesion to other cells or to ECM, and to enter tissue from vessels, the cells must resist flow and barriers, and must undergo transient interactions. Thus, force generation is required for motility, for reaching a distant target tissue. The formation of actin–myosin complexes allows ATP hydrolysis–dependent contractions, facilitating changes in cell shape and cell movement (Vale & Milligan, 2000; O'Connell et al, 2007). Our data implicate SWAP-70 in the control of these processes as we see increased cortical actin and myosin in KO cells, fitting the notion of their enhanced stiffness (Hosseini et al, 2020b). In cells expressing SWAP-70 ABM, we saw an increase in the ratio of cortical versus cytoplasmic actin and myosin even beyond that of KO cells, indicating that the SWAP-70 actin binding mutant may cause a dominant negative phenotype. This would be consistent with the significantly lower number of SWAP-70 ABM-expressing tumor cells accumulating in the lung.

RT-DC experiments indicate a lower apparent Young's modulus of SWAP-70–deficient cells. The difference between data seen in the AFM and RT-DC systems can be explained by the different timescales and the type of measurements used in the two approaches. Firstly, cells are repeatedly deformed at 1 Hz to measure the properties of a cell by AFM, whereas in RT-DC, the cells deform within milliseconds. Different strain rates were previously shown to affect relative contributions of the F-actin cytoskeleton and other cellular elements to the cell deformation response (Urbanska et al, 2020). Also, the cells might adapt to the forces they experience during the oscillatory AFM indentation. Second, RT-DC analyzes detached cells in a contactless manner, whereas in the AFM assay, round cells are probed while sitting on a surface. In any case, the important role of SWAP-70 was apparent in both systems. The rescue experiments showed that effects on cortical stiffness seen with both methods can be reversed by re-expressing SWAP-70 in the KO cells, indicating that SWAP-70 indeed contributes to this aspect of cellular mechanics.

Cell motility is a coordinated process that involves the formation of protrusive structures at the front pole and a dynamic retraction at the rear pole. Polymerization of actin filaments in lamellipodia and interaction of preformed actin filaments with myosin molecules in contractile bundles contribute to these processes, respectively. In such context, we observed morphological differences between control and SWAP-70–deficient cells, including a more prominent trailing tail in the latter, suggesting defects in its retraction—a source of magnupodium-like structures behind the migrating cells—and changes in the lamellipodium at the leading edge (Mejillano et al, 2004; Bemmerlein et al, 2022), impacting two characteristic features that contribute to cell motility with a mesenchymal (lamellipodial) migration mode (Shafqat-Abbasi et al, 2016). The presence of fewer membrane ruffles in KO cells is consistent with disruption of such active cell motility by the absence of SWAP-70 in agreement with other studies (Shinohara et al, 2002; Hilpela et al, 2003; Fukui & Ihara, 2010; Chacon-Martinez et al, 2013).

The latter phenotype could also be related to SWAP-70–dependent macropinocytosis and phagocytosis (Oberbanscheidt et al, 2007; Baranov et al, 2016), which require dynamic reorganization of the actin cytoskeleton and contractile activities, processes important for cancer cell invasion (Montcourrier et al, 1994). Furthermore, it cannot be excluded that the SWAP-70–dependent multicellular colony-like islets observed in Ctrl cells treated with TGFβ/EGF are also involved in collective invasive migration, similar to a role of E-cadherin previously suggested for this cell line (Elisha et al, 2018).

SWAP-70 binds to filamentous but not globular actin and bundles F-actin. Therefore, SWAP-70 promotes generation of more stable, stronger filaments (Chacon-Martinez et al, 2013), which are key to building larger cellular structures such as lamellipodia or F-actin rings (Pearce et al, 2006; Garbe et al, 2012; Roscher et al, 2016). As we showed earlier, SWAP-70, likely through its F-actin bundling activity, delays depolymerization and thus stabilizes actin filaments. In this report, we demonstrate that the interaction of SWAP-70 with F-actin contributes substantially to tumor metastasis. We suggest that the multifold effects of SWAP-70 promoting F-actin stabilization and thus changes in its dynamics support tumor cell metastasis and thus define a novel metastasis-promoting mechanism. The failure of a SWAP-70 mutant protein to bind F-actin to support several cell activities considered important for tumor cell metastasis supports this hypothesis. The F-actin binding domain is critical for cell migration, adhesion, and mechanics and supports tumor cell homing into the lung.

The altered biophysical and cellular properties of SWAP-70–deficient tumor cells correlate with reduced metastasis in mice. To examine metastasis in two very different tissues, we choose lung and bone, where typically metastases appear earlier in the lung than in the bone. For both target tissues, metastasis was several-fold reduced. In addition to the orthotopic breast cancer metastasis model done in the BALB/c strain of *Mus musculus*, we also used a metastasis model in a distinct mouse strain, and ectopic injection of Lewis lung cancer cells to provoke metastasis in C57BL/6 lungs. The results agree that metastasis is strongly reduced in both systems. Because lung and bone microenvironments including extracellular matrix are different, and yet, in both tissues, metastasis of SWAP-70–deficient tumor cells is reduced, this suggests a rather general metastasis-reducing feature of SWAP-70 deficiency.

Formation of metastatic tumors from a primary tumor requires several steps, from dissociation of cells from the primary tumor via entering vessels, traveling in vessels, exiting vessels, to invasion into a distant tissue and homing and expansion therein. Tumor cells may fail at any of these steps, all of which require dynamic adaptations of the cytoskeleton. To assess whether SWAP-70 is specifically required for the initiation of metastasis or is necessary at subsequent steps as well, we undertook tail vein injections of tumor cells to bypass the initial steps of dissociation from the primary tumor and entry into blood vessels. Clearly, SWAP-70 is also required for efficient metastasis in the later steps, that is, from the presence of tumor cells in the bloodstream to establishing metastases. Investigating extravasation of tumor cells from the blood vessel into the tissue revealed that without SWAP-70, the tumor cells are much reduced in attempting to cross the vessel wall. This combined with their altered mechanic properties and unbalanced cytoskeletal dynamics may make them more vulnerable, for

example, to the attack by the immune system (Lei et al, 2021; Liu et al, 2021) resulting in reduced establishment of metastatic tumors, be it inside or outside vessels. With the identification of SWAP-70 as a metastasis-promoting protein, therapeutic approaches targeting this protein seem an attractive perspective.

# Materials and Methods

### Cell culture and growth factor induction

The mouse breast carcinoma cell line 4T1 (ATCC CRL-2539) and the mouse LLC (ATCC CRL-1642) were purchased from the American Type Culture Collection (ATCC). 4T1 and LLC cell lines were cultured in DMEM (Invitrogen) supplemented with 10% (vol/vol) heat-inactivated FCS (Invitrogen) and 1% (vol/vol) penicillin and streptomycin (Pen-Strep; Invitrogen) (referred to as complete DMEM subsequently), under humidified conditions at 37°C and 5% $CO_2$. Cells were grown to ~70% confluence and split by trypsinization using 0.05% trypsin, EDTA solution (Invitrogen). For growth factor stimulation, $2 \times 10^5$ cells were seeded in a 10-cm petri dish overnight to let the cells attach and spread. The next day, the medium was removed, the cells were washed with 2 ml PBS, and the serum-depleted medium was added to starve the cells for 16 h (starvation medium). Then, cells were treated with complete medium containing 5 ng/ml TGFβ (7666-MB-005; R&D Systems) and 50 ng/ml EGF (236-EG-200; R&D Systems) for at least 48 h. Unless otherwise stated, 5 ng/ml TGFβ and 50 ng/ml EGF were used to stimulate cells.

### Generating KO mouse cancer cell lines

The CRISPR/Cas9 system was used to establish *Swap70* KO cells. Guide RNAs were chosen by the lowest probability of off-target effects (Guide 1 AGTGGCGCGAGCTGGACCTG, Guide 2 GCGAGCTG-GACCTGGCGTCG). The guides were cloned into the px458 plasmid, which codes for Cas9-GFP and the guide RNA. Transfection of the cancer cells was done using Lipofectamine 3000 (Invitrogen) transfection. After 24 h, GFP-positive cells were single cell–sorted into 96-well plates with Aria II (BD Biosciences). Cells were expanded and analyzed for the absence of SWAP-70 protein by immunoblotting, and mutation of exon 1 was confirmed by DNA sequence analysis.

### Immunoblotting

Total protein was harvested from cells lysed in RIPA buffer (50 mM Tris, pH 8.0, 150 mM NaCl, 0.5% sodium deoxycholate, 1% NP-40, 0.1% SDS, 5 mM NaOV4, 10 mM NaF) containing the complete protease inhibitor cocktail (Roche). The concentration of protein was measured using the Bradford protein assay (RotiQuant; Roth) according to the manufacturer's protocol. 10 μg of total protein from each sample was loaded into 8% or 10% sodium dodecyl sulfate–polyacrylamide gel electrophoresis (SDS–PAGE) for protein separation. Proteins in the gel were transferred to a polyvinylidene difluoride membrane using the semi-dry transfer cell (Bio-Rad) at 18 V for 45 min. The membranes were blocked by Tris-buffered

saline with PBS, 0.1% (vol/vol) Tween-20 (PBST), and 5% (wt/vol) milk powder for 30 min at RT, and then, the membranes were washed with PBST and incubated with anti-SWAP70 at 1 $\mu$g/ml (Borggrefe et al, 1998) O/N at 4°C. Washed by PBST, the polyvinylidene difluoride membrane was incubated with goat anti-rabbit HRP-coupled antibody for 1 h at RT. The protein was detected using enhanced chemiluminescence kits (Millipore Immobilon Western HRP Substrate) and the membrane subsequently developed on Amersham Imager 600 (GE Healthcare). Membranes were subsequently stripped and probed with anti-tubulin as a loading control.

### Transduction of 4T1 cancer cell lines

Retroviral plasmids expressing either GFP, a SWAP70-GFP fusion protein, or a SWAP-70 ABM-GFP fusion protein (Ocana-Morgner et al, 2013) were transfected into the packaging cell line Plat-E cells using polyethyleneimine (PEI) (PolyScience). 8 $\mu$g of DNA was mixed with 20 $\mu$g of PEI and 500 $\mu$l of DMEM without additives. The mix was vortexed and incubated for 20 min at room temperature. The mix was added dropwise to the Plat-E cells in a 10-cm petri dish containing 9.5 ml of additive-free DMEM. After 6 h, the mix was replaced with 10 ml of complete DMEM. After 48 h, the viral supernatant produced by the Plat-E cells was collected and filtered through a 0.45-$\mu$m filter (Sarstedt). The viral supernatant was then used to spin-infect 4T1 cells in the presence of 8 $\mu$g/ml of polybrene (Merck/Sigma-Aldrich). The viral supernatant was replaced with fresh complete DMEM the next day. The transduced cells were either be sorted by FACS or selected by antibiotics afterward.

### Mice

BALB/c mice were purchased from Janvier Labs. All animals were bred and maintained in the animal facility of the Medizinisch-Theoretisches Zentrum, Medical Faculty Carl Gustav Carus, Technische Universität Dresden (TUD). All animal experiments were done according to the national guidelines of animal welfare regulations and authorized by permit by the Landesdirektion Dresden.

### Orthotopic breast tumor model

5 × 10$^5$ 4T1 cells were suspended in 100 $\mu$l PBS and mixed with 100 $\mu$l stock Matrigel (356231; BD Biosciences) (1:1 mixture). A syringe with a 26G needle was used to inject the Matrigel cell suspension directly into the fourth mammary gland of female 8- to 12-wk-old BALB/c mice under anesthesia. Before injection, the Matrigel and the Matrigel cell suspension should stay on ice to avoid gelification. Primary tumors and mouse health were regularly controlled, length (L) and width (W) were measured weekly using calipers, and tumor volume (V) was calculated as V = (L × W2)/2. Lungs and both hind bones were collected to determine the number of metastatic tumors. A dissecting microscopy was used to count the visible metastatic nodules on the lungs defined here as the macrometastasis. Hind bones from each mouse were analyzed by micro-computed tomography ($\mu$CT) scanning, and a 3D construct was produced to visualize bone lesions. The resulting 3D construct was used to classify bones into three categories (from I, least

severe, to III, the most severe) depending on the degree and the number of bone erosions. Primary tumor, lung, and bone samples were then fixed by formalin, paraffin-embedded, and then sectioned on a microtome. Sections were stained with hematoxylin and eosin (H&E), and tumors were identified microscopically.

### Measurement of short-term accumulation of 4T1 cells in lungs and extravasation

1 × 10$^6$ GFP-expressing 4T1 cells were injected intravenously into the 8- to 12-wk-old female BALB/c mice. For the analysis of cells expressing either the wild-type SWAP-70 or the SWAP-70 ABM, rather than using GFP as a marker, cells were labeled with 2 $\mu$M CMFDA (Thermo Fisher Scientific) immediately before injection. The mice were euthanized 16 h post-injection, and the lungs were collected to quantify GFP- or CMFDA-positive 4T1 cells by flow cytometry. To isolate and quantify GFP cells, lungs were minced into small pieces in a petri dish and treated with digestion enzymes (0.25 U/ml dispase [BD Biosciences], collagenase [Merck/Sigma-Aldrich], 100 $\mu$g/ml and 7.5 $\mu$g/ml DNase [Merck/Sigma-Aldrich] in 9 ml of PBS) for 1 h at 37°C. The resulting suspension was then passed through a 100-$\mu$m cell strainer and centrifuged at 400$g$ for 5 min. The cell pellet was resuspended in 1 ml of ammonium–calcium–potassium lysis buffer on ice for 5 min to lyse red blood cells. Lysis was stopped through the addition of 5 ml cold FACS buffer, and the cells were then collected by centrifugation. The resulting cell pellet was resuspended in FACS buffer and centrifuged once more, after which the cells resuspended in FACS buffer were analyzed by flow cytometry.

To investigate the ability of the tumor cells to extravasate from the blood vessels into the surrounding lung tissue, 5 × 10$^5$ CMFDA-labeled 4T1 cells were injected as above. The mice were euthanized after 18 h, and their lungs were inflated with 1:1 OCT:PBS and frozen in OCT at –80°C. Three experiments, with each two mice (Ctrl or G3KO), were performed. Frozen lungs were sectioned at 18 $\mu$m width. The sections were fixed with 4% polyformaldehyde in PBS for 10 min at RT, permeabilized with 0.01% Triton X-100 for 5 min, and incubated in 5% BSA in 0.01% Triton X-100/PBS at RT. Primary anti-CD31 antibody (#550274; BD Pharmingen)was used at 1:100 dilution and incubated overnight at 4°C, whereas secondary antibody (#712-166-153; Code) was diluted 1:1,000 in blocking solution and incubated for 1–2 h at RT with the sample. After each antibody incubation, the sections were washed 5x for 3 min each in PBS; mounting medium and cover glass were applied to the sections. The samples were imaged with Leica STELLARIS 8 at 43x. At least two regions per mouse (sections separated at least 100 $\mu$m) of size 1,675.91 × 1,675.91 $\mu$m (2316 × 2316 pixels) were scanned, and all of the CMFDA-labeled cells were imaged using Z-scanning (0.3-$\mu$m steps) at 5x magnification with 256 × 256 pixel resolution. For statistical analysis, the unpaired $t$ test was used.

### Soft agar colony formation assays

1 ml of 0.5% (wt/vol) agar in RPMI tissue culture media was added to each well of a six-well plate and allowed to solidify to create a base. 3 × 10$^4$ cells were mixed with agarose (final concentration of agarose 0.35% [wt/vol] in RPMI), and 300 $\mu$l of the mixture was

layered onto the base. The complete cell culture medium (with or without TGFβ/EGF) was added on top of the agarose and changed every 3 d. After two wk, the cells were fixed with 10% methanol and 10% acetic acid in ddH2O for 10 min and colonies were stained with 0.01% crystal violet for 1 h. Crystal violet was removed, the samples were washed several times with ddH2O until the stained single colonies were clearly visible, then the wells were scanned using an Epson Perfection 4180 scanner, and colonies were counted with Fiji software.

### Wound-healing assay

Flat glass-bottom 96-well plates were precoated with fibronectin (bovine plasma fibronectin; Merck/Sigma-Aldrich), which had been resuspended in PBS with 1 mM MgCl$_2$ and 1 mM CaCl$_2$ at a concentration of 25 µg/ml for 2 h at 37°C. The Oris (Platypus Technologies) silicon stopper was placed into the 96-well fibronectin-coated plate to create a physical barrier, and 5 × 10$^4$ cells were seeded around the stopper with complete cell medium with, or without, TGFβ/EGF overnight at 37°C. After removing the stoppers, migration of the cells into the resulting void was imaged for each sample every 30 min for up to 15 h on a Nikon TE2000E inverted microscope with a Plan APO 4x/0.13 NA objective, a CoolSNAP HQ camera (Photometrics), and standard filter sets at 37°C. All images were acquired using the Nikon NIS-Elements ND2 software and analyzed with Fiji.

### Transwell (Boyden chamber) cell migration assay

5 × 10$^4$ cells in 100 µl starvation medium were added to the upper chamber of a Transwell insert (6.5 mm diameter, 8 µm pore size; Corning), and the insert was placed in a 24-well plate containing 1 ml complete medium with or without TGFβ/EGF. The Transwell inserts were fixed with 4% PFA after 24 h. Cells that remained on the upper side of the Transwell were removed with a cotton swab, whereas cells successfully migrated to the bottom of the Transwell were stained with the DAPI staining solution. Three randomly selected fields on the lower side of the insert were imaged with an Olympus IX 70 inverted microscope at 10x magnification, and the number of cells in each field was then counted using Fiji software. The average number of cells per insert was calculated and normalized to the control.

### Inverted Transwell (Boyden chamber) cell invasion assay

Transwell inserts (6.5 mm diameter, 8 µm pore size; Corning) were coated on the top with 100 µl growth factor–reduced Matrigel (356231; BD Biosciences). 5 × 10$^4$ cells were seeded on the bottom of the Transwell inserts and allowed to adhere for 4 h. Transwell inserts were inverted, and the medium containing 10% FCS and TGFβ/EGF was added to the top as the chemoattractant, and in the lower compartment, the medium containing just 0.5% FCS was added. Cells were incubated for 3 d and then stained with 10 µM CellTracker CMFDA dye (C2925; Thermo Fisher Scientific) for 1 h at 37°C. Invaded cells that crossed the membrane pore and invaded the Matrigel were imaged by confocal microscopy (Zeiss LSM 880) with a 20x objective with optical Z-sections scanned at 2-µm

intervals moving up from the underside of the membrane into the Matrigel. The results were quantified from three randomly chosen views from each insert.

### Cell adhesion assay on fibronectin-coated plates

Flat glass-bottom 96-well plates were precoated with fibronectin (as in the 2D migration assay). 1 × 10$^6$ cells/ml were labeled with 1 µM CellTracker CMFDA dye (C2925; Thermo Fisher Scientific) for 40 min at 37°C. The labeled cells were washed and resuspended in HBS solution (0.14 M NaCl, 5 mM KCl, 1 mM CaCl$_2$, 0.4 mM MgSO$_4$, 0.5 mM MgCl$_2$, 0.3 mM Na$_2$HPO4, 6 mM glucose, 4 mM NaHCO$_3$), and 3 × 10$^4$ cells in 100 µl were added in each well of the 96-well fibronectin-coated plate. Plates were incubated for 30 min at 37°C to let the cells adhere and then carefully washed twice with PBS to remove the non-adherent cells. Adherent cells were fixed in 100% methanol for 10 min and visualized with an Olympus IX 70 inverted microscope at 10x magnification. Samples were analyzed in triplicates, and three fields of view were imaged per well. Cells per field of view were quantified using Fiji software.

### Cell adhesion assay on frozen mouse lung section

Mouse lungs were placed in OCT compound (Tissue-Tek 4583), snap-frozen, and kept at −80°C until the experiment. Lung cryosections 8 µm thick were placed on glass slides. A chamber was created by gluing the neck, cut from a 1.5-ml micro-centrifuge tube, around the lung section. Cells were labeled with CellTracker CMFDA dye (C2925; Thermo Fisher Scientific), for 40 min at 37°C, washed, and resuspended in HBS solution, and 0.25 × 10$^6$ cells in 500 µl were added to each chamber. Glass slides with cells were placed on a shaker and slowly agitated for 30 min at room temperature. Non-adherent cells were removed by washing four times with PBS. Adherent cells were fixed in 100% methanol for 10 min and imaged with an Olympus IX 70 inverted microscope at 10x magnification, and two to three fields of view were imaged per lung section. Each sample was analyzed in duplicates. Fiji software was used to quantify the number of cells and the lung area per field of view. The average number of cells per unit lung area was calculated and normalized to the control sample.

### Quantification of membrane ruffles

3 × 10$^5$ 4T1 cells were seeded in 10-cm dishes and allowed to settle overnight. The next day, the cells were starved ON followed by stimulation with TGFβ/EGF for 48 h. Stimulated cells were seeded on fibronectin-coated cover glasses and incubated overnight in the medium containing TGFβ/EGF. Cells were fixed with 4% PFA, permeabilized in 0.1% Triton X-100 in PBS, blocked in 1% BSA, and stained with 2.5 U/ml phalloidin and 0.1% DAPI in 1% BSA in PBS. Eight fields of view within three cover glasses per sample with an average of 32 cells per field were acquired on an Olympus IX70 wide-field microscope at 20x magnification. Analysis was performed in Fiji: the total number of cells per field of view was counted by the DAPI signal, the percentage of the cells having at least one ruffle was calculated, and the t test was applied.

## Zymography and matrix degradation assays

For zymography analysis of matrix metalloprotease activity, 20 $\mu$l of conditioned cell culture medium from unstimulated and stimulated 4T1 cells grown to confluency was mixed with 5 $\mu$l of SDS–PAGE loading buffer containing 10% SDS, 4% sucrose, and 0.1% bromophenol blue; 7 $\mu$l of the sample was loaded into 7.5% SDS–polyacrylamide gels containing 0.05% gelatin. Gels were incubated for 30 min in 2.5% Triton X-100 followed by overnight incubation in activating buffer (0.05 M Tris, pH 7.4, 0.005 mM CaCl2, 0.001 mM ZnCl2). Gels were stained for 30 min in Coomassie blue and destained with 80% acetic acid, 10% glycerol, 10% methanol. Gelatin degradation seen as light bands was analyzed using Fiji software.

To test matrix degradation by tumor cells, gelatin-coated coverslips were prepared according to the protocol by Martin et al (2012). Briefly, 12-mm coverslips were cleaned in 20% nitric acid, coated with 50 $\mu$g/ml poly-L-lysine, treated with 0.5% glutaraldehyde solution, and covered with 1:8 mixture of FITC-conjugated gelatin (M1303-5; BioVision) and 5% (w/w) sucrose/unlabeled gelatin solution for 20 min at 4°C. Gelatin-coated slides were placed in 5 mg/ml sodium borohydride solution, washed in PBS, sterilized in ethanol, further washed, and equilibrated in cell culture medium for 1 h in 24-well plates. Unstimulated and stimulated 4T1 cells at 10 × 10$^3$ cells per well were added to the coverslip and kept at 37°C in the cell culture incubator for 48–72 h. Cells were fixed in 4% PFA and stained with DAPI. The slides were mounted with Vectashield mounting medium and imaged on an Olympus IX70 wide-field inverted microscope using a 100x oil immersion objective. The area of the degraded gelatin and the number of nuclei per field of view were analyzed using Fiji software.

## SEM

4T1 cells cultured in DMEM as described above with or without TGF$\beta$/EGF treatment were plated on the fibronectin-coated glass coverslips for 24 or 48 h before processing for SEM as described previously (Bemmerlein et al, 2022). To that end, cells were fixed in glutaraldehyde (2%) for 1 h at RT and then put at 4°C overnight. After 2-h post-fixation in osmium tetroxide (1%) at 4°C, samples were subjected to dehydration in an acetone gradient (25–100%) and critical point–dried in a $CO_2$ system (Critical Point Dryer, EM CPD 300; Leica Microsystems). Afterward, samples were sputter-coated with gold (sputter-coating device SCD 050; BAL-TEC GmbH) and observed at a 5-kV accelerating voltage with a field emission scanning electron microscope (JSM 7500F; Jeol).

## AFM setup and measurements

For AFM measurements of suspended cells, cell culture dishes (FluoroDish FD35-100) and wedged cantilevers were plasma-cleaned for 2 min and then coated by incubating the dish at 37°C with 0.5 mg/ml poly-L-lysine–polyethylene glycol dissolved in PBS for at least 1 h (poly-L-lysine(20)–g[3.5]–polyethylene glycol(2); SuSoS) to prevent cell adhesion. Cells were collected by trypsinization, washed, and resuspended in $CO_2$-independent DMEM (PN: 12800-017; Invitrogen) with 4 mM NaHCO3 buffered with 20 $\mu$M Hepes/NaOH (pH 7.2), for AFM experiments ~2 h before the

measurement (Stewart et al, 2011; Fischer-Friedrich et al, 2014; Fischer-Friedrich et al, 2016; Hosseini et al, 2020b).

The experimental setup included an AFM (NanoWizard I) that was mounted on a Zeiss Axiovert 200M optical, wide-field microscope using a 20× objective. Cell culture dishes were kept in a Petri dish heater (JPK Instruments) at 37°C during the experiment. Before every experiment, the spring constant of the cantilever was calibrated by thermal noise analysis (built-in software; JPK) using a correction factor of 0.817 for rectangular cantilevers. The cantilevers used were tipless, 200–350 $\mu$m long, 35 $\mu$m wide, and 2 $\mu$m thick (CSC37, tipless, no aluminum; MikroMasch). The nominal force constants of the cantilevers ranged between 0.2 and 0.4 N/m. The cantilevers were supplied with a wedge, consisting of UV curing adhesive (Norland 63; Norland Products) to correct for the 10° tilt (Stewart et al, 2013). The measured force, piezo height, and time were output with a time resolution of 500 Hz.

Preceding every cell compression, the AFM cantilever was lowered to the dish bottom in the vicinity of the cell until it touched the surface and then retracted to ≈14 $\mu$m above the surface. Subsequently, the free cantilever was moved and placed on top of the cell. Thereupon, a brightfield image of the equatorial plane of the confined cell was recorded to evaluate the equatorial radius Req at a defined cell height h. Cells were confined between the dish bottom and the cantilever wedge. Then, oscillatory height modulations of the AFM cantilever were carried out with oscillation amplitudes of 0.25 $\mu$m at a frequency of 1 Hz. During this procedure, the cell was on average kept at a normalized height h/D between 60% and 70%, where D = 2(3/(4$\pi$)V)1/3 and V is the estimated cell volume. For each experiment, different conditions were measured on the same day with the same average cell-confinement height and the same AFM cantilever.

The data analysis procedure was described earlier in detail (Fischer-Friedrich et al, 2016). In our analysis, the force response of the cell is translated into an effective cortical tension $\gamma$ = F/[Acon(1/R1 + 1/R2)], where Acon is the contact area between the confined cell and AFM cantilever, and R1 and R2 are the radii of principal curvatures of the free surface of the confined cell (Fischer-Friedrich et al, 2014; Fischer-Friedrich et al, 2016; Hosseini et al, 2021). Oscillatory force and cantilever height readouts were analyzed in the following way: for every time point, effective cortical tension $\gamma$eff and surface area strain $\varepsilon$A(t) = (A(t) − ⟨A⟩)/⟨A⟩ were estimated. An amplitude and a phase angle associated with the oscillatory time variation of effective tension and surface area strain are extracted by sinusoidal fits. To estimate the value of the complex elastic modulus at a distinct frequency, we determine the phase angles and amplitudes, active cortical tension, and surface area strain, respectively. The complex elastic modulus at this frequency is then calculated as A$\gamma$/A$\varepsilon$ exp(i($\varphi\gamma$ − $\varphi\varepsilon$)).

Statistical analyses of cortex mechanical parameters were performed in MATLAB using the commands "boxplot" and "ranksum" to generate boxplots and determine $P$-values from a Mann–Whitney $U$ test (two-tailed), respectively.

## Transfecting and quantitative analysis of cortex-associated actin and myosin II

Lipofectamine 3000 (Invitrogen) was used for transfecting plasmids into the cell lines according to the manufacturer's recommendations.

The MApple-LC-Myosin-N-7 (plasmid 54920; Addgene; http://n2t.net/addgene: 54920; RRID: Addgene_54920) and the MCherry-Actin-C-18 plasmids were gifts from Michael Davidson (plasmid 54967; Addgene; http://n2t.net/addgene: 54967; RRID: Addgene_54967). For imaging, the cells were transferred to PLL-g-PEG–coated FluoroDishes (FD35-100) with $CO_2$-independent culture medium (described previously). During imaging, they were maintained at 37°C, and an image was taken of the equator (defined as the largest cross-sectional area of the cell) using Leica TCS SP5 and used to analyze the equatorial region. For every image, the pixel size, laser power, and gain were fixed. For quantitative analysis of cortex-associated actin and myosin II, the images were analyzed as described previously (Hosseini et al, 2020b; Hosseini et al, 2022). Briefly, using a MATLAB custom code, the cell boundary was identified. Along this cell boundary, 200 radial, equidistant lines were determined by extending 1 $\mu m$ to the cell interior and 1 $\mu m$ into the exterior. The radial fluorescence profiles corresponding to these lines were averaged over all 200 lines. This averaged intensity profile is then fitted by a function described in Hosseini et al (2020b). The 2D normalized cortical and cytoplasmic fluorescence intensities were obtained and used to measure the cortex-to-cytoplasm ratio in the boxplots.

### Real-time deformability cytometry

As described in Bemmerlein et al (2022), cells were either non- or stimulated with growth factors for 48 h. Subconfluent 4T1 cells were trypsinized, washed, and resuspended in 0.5% methylcellulose/PBS with 50 $\mu g$/ml DNase (>2.000 U/mg protein; Sigma-Aldrich). The real-time deformability cytometry (RT-DC) measurements were performed as previously described (Otto et al, 2015). Briefly, about 30 $\mu l$ of cell suspension was loaded into a 1-ml syringe and run through a microfluidic device with a 30-$\mu m$-wide channel (Zellmechanik Dresden) at a flow rate (sheath and sample fluid) of 0.16 and 0.32 $\mu l$/s. At the end of the microfluidic channel, images of cells were captured by a high-speed camera and cell size (projected area) and deformation (D = 1-circularity) were automatically detected using shape-in software (version 2.0.8; Zellmechanik Dresden). The peak fluorescence signal was simultaneously obtained for each analyzed cell (488-nm excitation, 525/50-nm emission). GFP-positive cells were analyzed after gating for the fluorescence–high cell fraction. Moreover, a brightfield image was acquired for every measured cell, making the data available for multiparametric offline analysis. Data analysis and calculation of the apparent elastic modulus were performed in ShapeOut 1.0.6 (available at https://github.com/). In each independent experiment (n = 5), 10,000 cells were measured per cell type/condition. Experiments were conducted at RT.

### FACS analysis

Cells were serum-starved for 16 h and then cultured in complete cell culture media (10% FCS) in the presence or absence of TGF$\beta$/EGF for 48 h and then collected by trypsinization. They were then washed and resuspended in FACS buffer of EMT marker staining together with BV-421 anti-EpCAM (11825; BioLegend), FITC anti-CD106 (105705; BioLegend), AF647 anti-CD61 (104313; BioLegend), and PE anti-CD51 (104105; BioLegend). For apoptosis staining, cells were resuspended in Annexin V binding buffer and stained according to the manufacturer's protocol with Annexin V (BioLegend) and then DAPI. FACS analysis was then performed for all samples on a FACSymphony A3 machine (BD).

### qRT–PCR analysis

Cells were grown to ~70% confluence, then serum-starved overnight before stimulation with TGF$\beta$/EGF for the times indicated. Total RNA was then isolated using an RNeasy Mini kit (QIAGEN). cDNA was then generated from 2 $\mu g$ of RNA using oligo(dT)$_{15}$ primers (Promega) and M-MLV reverse transcriptase (Promega). qRT–PCR was performed with a QuantiNova SYBR Green PCR kit (QIAGEN) using primers against *Snail2* (Fwd TCCCATTAGTGACGAAGA, Rev CCCAGGCTCACA-TATTCC), *Twist1* (Fwd AGCGGGTCATGGCTAACG, Rev CCCAGGCTCACA-TATTCC), and *Gapdh* (Fwd TGAAGGTCGGTGTGAACGGATTTGGC, Rev CATGTAGGCCATGAGGTCCACCAC). Amplification was performed on a qTOWER2 (Analytik Jena). The $C_t$ value of *Gapdh* was used as a reference to calculate the $\Delta C_t$. To compare the change in mRNA expression after induction within the populations, $\Delta C_t$ of the 0-h time point was used as a reference to calculate the $\Delta\Delta C_t$ for the 12- and 24-h time points.

### Transcriptomics

Cells were FACS-sorted into tubes containing lysis buffer for RNA isolation, with 300 cells being analyzed per experiment, n = 3 experiments. The sequencing was performed on a HiSeq 2500 (Illumina, Inc.) sequencer by Deep Sequencing Facility (Center for Molecular and Cellular Bioengineering, TUD, Dresden). Transcriptome data were further analyzed using online Metascape (Zhou et al, 2019), GSEA (Mootha et al, 2003; Subramanian et al, 2005), and Ingenuity Pathway Analysis (QIAGEN) software.

## Supplementary Information

## Acknowledgements

We thank the Electron Microscopy/Histology (EMH) Facility and the Deep Sequencing Facility at the Technology Platform of the Center for Molecular and Cellular Bioengineering (TUD) for instrument access and support. M Rauner was supported by the Deutsche Forschungsgemeinschaft (SPP2084, RA1923/22-1, and RA1923/14-2); E Fischer-Friedrich was supported by the Deutsche Forschungsgemeinschaft within Germany's Excellence Initiative, EXC-2068-390729961, Cluster of Excellence Physics of Life of TU Dresden, the Heisenberg Program 495224622 (FI 2260/8-1), and the grant FI 2260/7-1; D Corbeil was supported by the Deutsche Forschungsgemeinschaft (SPP2084, CO 298/11-1); S Perner was supported by the Deutsche Forschungsgemeinschaft (SPP2084, project number 401179983); A Taubenberger was supported by the Deutsche Krebshilfe (MSNZ); and R Jessberger was supported by the Deutsche Forschungsgemeinschaft (SPP2084, JE150/29-2), and by a grant from Thorne Inc.

## Author Contributions

C-Y Chang: conceptualization, data curation, formal analysis, validation, investigation, visualization, methodology, and writing—review and editing.
G Pearce: conceptualization, data curation, formal analysis, validation, investigation, methodology, and writing—review and editing.
V Betaneli: data curation, formal analysis, validation, investigation, visualization, methodology, and writing—review and editing.
T Kapustsenka: data curation, formal analysis, investigation, and writing—review and editing.
K Hosseini: data curation, formal analysis, investigation, visualization, and methodology.
E Fischer-Friedrich: data curation, formal analysis, funding acquisition, validation, investigation, visualization, methodology, and writing—review and editing.
D Corbeil: data curation, formal analysis, validation, investigation, visualization, methodology, and writing—review and editing.
J Karbanová: data curation, formal analysis, investigation, visualization, and methodology.
A Taubenberger: data curation, formal analysis, investigation, visualization, methodology, and writing—review and editing.
B Dahncke: data curation, formal analysis, validation, investigation, visualization, and methodology.
M Rauner: data curation, formal analysis, validation, methodology, and writing—review and editing.
G Furesi: data curation, investigation, and methodology.
S Perner: investigation, visualization, methodology, and writing—review and editing.
F Rost: software, formal analysis, validation, visualization, methodology, and writing—review and editing.
R Jessberger: conceptualization, data curation, formal analysis, supervision, funding acquisition, validation, methodology, project administration, and writing—original draft, review, and editing.

## Conflict of Interest Statement

The authors declare that they have no conflict of interest.

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
