## [Reviewer comments · Life Science Alliance]

Life Science Alliance

The F-actin bundler SWAP-70 promotes tumor metastasis

Chao-Yuan Chang, Glen Pearce, Viktoria Betaneli, Tatsiana Kapustsenka, Kamran Hosseini, Elisabeth Fischer-Friedrich, Denis Corbeil, Jana Karbanova, Anna Taubenberger, Björn Dahncke, Martina Rauner, Giulia Furesi, Sven Perner, Fabian Rost, and Rolf Jessberger

DOI: <https://doi.org/10.26508/lsa.202302307>

Corresponding author(s): Rolf Jessberger, TU Dresden

Review Timeline:

Submission Date:	2023-08-07
Editorial Decision:	2023-11-03
Revision Received:	2024-05-02
Editorial Decision:	2024-05-06
Revision Received:	2024-05-08
Accepted:	2024-05-09

Transaction Report:

November 3, 2023

Re: Life Science Alliance manuscript #LSA-2023-02307

Prof. Rolf Jessberger
TU Dresden
Inst. of Physiological Chemistry
Fiedlerstr. 42
Dresden 1307
Germany

Dear Dr. Jessberger,

Thank you for submitting your manuscript entitled "The F-actin bundler SWAP-70 promotes tumor metastasis" to Life Science Alliance. The manuscript was assessed by expert reviewers, whose comments are appended to this letter. We invite you to submit a revised manuscript addressing the Reviewer comments.

Thank you for this interesting contribution to Life Science Alliance. We are looking forward to receiving your revised manuscript.

Sincerely,

B. MANUSCRIPT ORGANIZATION AND FORMATTING:

Reviewer #1 (Comments to the Authors (Required)):

Chang et al describe the role of SWAP-70 on tumor metastasis. The report provided demonstrates that SWAP-70 can affect tumor growth and metastasis, and plays a role in migration and invasion of cells. While at least in vitro it is shown that SWAP-70 F-actin interaction is important in its regulation of migration, connecting that back in vivo would provide full closure to the study.

Major comments-

1) Have the authors evaluated the effect of the SWAP-70 binding mutants in vivo to see if tumor growth/metastasis is still affected? This experiment is important to validate the claim that SWAP-70 interaction with F-actin is key to its effects.

Minor Comments-

1) A magnified inset for H&E staining would make it easier to see the lung metastasis.

2) Do you have H&E staining for the bone to see tumor cell infiltration?

3) It might be interesting to do a transcriptome analysis of the SWAP-70 actin binding mutant to determine what changes are due to SWAP-70 f-actin binding and which are independent of this binding.

Reviewer #2 (Comments to the Authors (Required)):

In this manuscript, Chang et al demonstrate a role of SWAP-70, an F-Actin bundling protein, in tumor progression, specifically linked to tumor migration, invasion and adhesion. The authors use different in vitro and in vivo systems that allow them to decipher the specific role of such protein, and to narrow down the possible stages of the metastatic cascade in which SWAP-70 is involved. As well, they use biophysical tools to attribute a role for this protein in cell mechanics, demonstrating that the lack of SWAP-70 increases cortical elements such as myosin or actin, which in turn increase cortical tension and stiffness, linking this with an increase ability of the cells to migrate and invade in in vitro systems, and to form metastasis in different mouse models of breast cancer.

Although the authors have performed a thoroughly analysis of the different stages in which SWAP-70 might be involved, I find that at the mechanistic level the role of this F-Actin bundler is not completely clear, and additional experiments are necessary in order to clarify certain aspects of the manuscript.

I have two major concerns related to the conclusions that the authors claim in the manuscript, related both to the particular stage of metastasis in which this protein might be implicated (i), and to the specific mechanism controlling it (ii):

(i): In light of the results, it looks clear that SWAP70 is not involved in cell growth (or not majorly as one of the KO cell lines have a significantly reduced ability to grow in mice models and the KO also affects colony formation in soft agar assays), nor in EMT, and mostly involved specially on invasion/adhesion/migration. However, the reduction in the number of mets observed in mice models could be due to only some of those steps, or to all at once. To decouple these factors in vivo, it would be nice to for example know if there is a difference in the number of circulating tumor cells in such models. This will already indicate if the intravasation would be impacted. Based on the tail vein injection experiments, it looks that the survival in the circulation/extravasation abilities/colonization is also affected, but the readout the authors chose for such experiments do not allow to discriminate between them all. The authors could perform the same experiments but fixing the lungs at a set time point(s), to understand whether the cells are stuck at capillaries or extravasated, or if they express apoptosis-related markers (cleaved-caspase 3 for instance). However, this reviewer is aware of the time and effort required for in vivo experiments.

As well, and taking advantage of the RNAseq already performed, it would be nice if the authors could show more of this data related to other cellular functions other than EMT. This could be useful for example to support the data related to changes in the

mechanical properties of cells upon KO. As well, related to the EMT phenotype, it would be nice to see the localization (nuclear/cytoplasmic) of some canonical EMT transcription factors to rule out changes associated to EMT, specially in light of the results of figure 3A/B, in which KO cells seem to be the only ones losing cell-cell contacts upon treatment.

(ii): Related to the mechanism that controls the effects of SWAP70 over tumor progression, the following questions should be addressed:

a) Most of the manuscript relies on the use of TGF β and EGF, which boost the ability of control cells (but not KO cells) to perform several metastasis-related functions. It would be nice to show if both the receptors of such factors, as well as some intracellular effectors (e.g SMADs for TGF β), are altered upon KO, to rule out that the effect that is observed in migration, invasion... could just be a lack of sensitivity to such factors.

b) One of the main claims of the authors in the manuscript is related to the fact that SWAP70 can control global cell stiffness, by altering cortical cytoplasmic elements that in turn modify cortical stiffness and tension. These changes in cell mechanics would change the ability of cells to squeeze through matrix pores thus altering their invasion abilities through changes in cell mechanics. Even if this might be true, the authors should rule out the possible changes in matrix degradation upon KO, to attribute such effects to a mechanically-linked phenotype.

c) The changes in cortical tension might be altering the cellular localization of mechanosensors like YAP, which could be regulating the expression of downstream genes. Performing YAP stainings in the different conditions (Control, KO and rescued phenotype) could help addressing this point. As well, and as one of the main effect that the authors find altered is adhesion, it would be important to check focal adhesion formation in the different conditions, which might be changed based on the images that are shown, and also because it is well known that tension is a regulator of focal adhesion maturation, which could be regulating adhesion in this context.

Minor points:

- A quantification of the supplementary figure 2D needs to be provided to support the linked claims.
- Although the authors already speculate about the possible differences (opposite) between AFM and RT-DC, checking actin/myosin in cells in the conditions that are used for RT-DC might be useful (although not necessary).
- In page 10, Supl Figure 5E is wrongly cited.

Chang et al., The F-actin bundler SWAP-70 promotes tumor metastasis

Response to Reviewers

We thank the reviewers for their insightful comments which helped us to substantially improve the manuscript.

The main changes to the manuscript are:

- inclusion of new *in vivo* data using a SWAP-70 actin binding mutant to further prove that also *in vivo* the actin binding property of the protein is key to its role in metastasis (Fig. 5H and Suppl. Fig. 6D)
- inclusion of high magnification images showing the H&E-stained lung metastasis as requested (Fig. 1D); we are also presenting higher magnification images of the bone metastasis (Suppl. Fig. 1B)
- inclusion of new data showing the analysis of formation of focal adhesions by control versus SWAP-70 deficient cells. The results show failure of SWAP-70 deficient cells to induce focal adhesions when stimulated (Fig. 2G).
- inclusion of data on extravasation of control or SWAP-70 deficient tumor cells from blood vessels in the lung. Quantification demonstrates that the mutant cells are less capable of extravasating (Fig. 1H and Suppl. Fig. 1F)
- inclusion of new data on matrix degradation and activity of matrix metallo proteinases, with both activities not different between control and SWAP-70 deficient cells (Suppl. Figs. 5B, 5C).

In the following we are responding to each of the points made by the reviewers. In the main text, all significant changes are typed in red.

Reviewer #1

1. While we had already shown that the actin binding feature of SWAP-70 is central to its metastasis-related functions in cell culture, we agree with the referee that it would be very good to show this also *in vivo*, as demanding as such experiments are. Thus, we have used SWAP-70 deficient 4T1 cells and re-expressed either the full-length SWAP-70 or the SWAP-70 actin binding mutant protein, which we have used also for biophysical experiments to allow direct comparison of *in vitro* with *in vivo* data. These cells were injected into the tail vein and their presence in the lungs 18 h later measured by FACS (new Fig. 5H and Suppl. Fig. 5C). The results show that the actin binding mutant not only fails to complement SWAP-70 deficiency but even seems to produce a mild dominant negative phenotype. Thus, consistent with the cell culture experiments (Fig. 5A, B, E, F), actin binding of SWAP-70 is required for tumor cells to accumulate in the lung.

Minor Points

1. We have added high magnification insets into Figure 1D showing the H&E-staining of lung metastasis as recommended by the referee.
2. We had already presented H&E staining for the bone sections (Suppl. Figure 1B) but apologize if that was not made sufficiently clear, which we think it is now. In addition we are showing high magnification insets of these H&E-stained bone sections.
3. Although we agree with the referee that transcriptome analyses would in principle be interesting, the problem is the large number of variations that one may consider (control

versus mutant cells, tumor cells in situ, different time points of metastasis, etc.). Given that SWAP-70 is not a transcription factor and thus any effects would be (highly) indirect, this adds to the complexity and difficulties in interpretation of the data. The effects on the transcriptome by a protein that modulates the F-actin cytoskeleton would be difficult to mechanistically track down to SWAP-70. Thus, we limited our transcriptome analysis to the EMT question. Therefore, while a potentially interesting avenue once the direct effects, processes and mechanics have been determined, a comprehensive transcriptome study cannot be within the scope of this paper.

Reviewer #2

In vivo experiments are indeed demanding and require long time periods (which sometimes include obtaining new permits to perform animal experiments, which may be impossible to obtain within a time frame reasonable for a revision). Nevertheless we succeeded to perform two types of additional in vivo experiments: extravasation and rescue by the SWAP-70 actin binding mutant.

(i) Fig. 1H and Supplemental Fig. 1D now show new data on extravasation of control or SWAP-70 deficient tumor cells from blood vessels into the surrounding tissue. In accordance to the literature (Al-Mehdi et al., 2020; PMID: 10613833), we saw little completely extravasated cells; 4T1 breast cancer cells tend to more often stay within lung vessels. Nevertheless, a fraction of control cells was extending protrusions into the tissue and are (attempting) extravasating 18 h after tail vein injection while none of the SWAP-70 deficient cells did (except for one single case, probably an outlier which we nevertheless show).

(ii)

a) We had checked expression levels of TGF β R and EGFR on the mRNA level and found them to be unaltered by SWAP-70 deficiency. EGFR surface expression was also assessed by FACS and found the same between control and the mutant cells.

Unfortunately a panel of available anti TGF β R antibodies did not work in FACS and thus we could not check this on that level. However, the SWAP-70 deficient cells reacted to EGF and TGF β the same as control cells with respect to proliferation, early and late apoptosis, expression of a series of EMT-related proteins (CD106, CD61, CD51, EpCAM measured by FACS) and of mRNAs (Snail, Twist), or in MMP activity and matrix degradation (Suppl. figures 2A-C, 5B, 5C).

Thus, there is no lack of sensitivity towards these two stimuli and no ubiquitous difference in their pathways between control and mutant cells. We are making this more clear now on page 9.

b) We thank the referee for asking for data on matrix degradation. We undertook two types of experiments in this regard: degradation of labeled gelatin matrix and zymography to check expression of matrix metallo proteinases (mostly MMP2 and MMP9). There was no difference between control and SWAP-70 deficient cells in expression of these proteases regardless whether the cells were stimulated or left untreated (new Suppl. Fig. 5B).

Likewise, there was no difference in matrix degradation, which was equally up-regulated upon stimulation of the control and SWAP-70 deficient cells (new Suppl. Fig. 5C).

c) We also thank the referee for suggesting the analysis of formation of focal adhesions as these are indeed linked to the underlying actin network. The new figure 2G shows data on both, the number of focal adhesions and the area the focal adhesions cover. While

before stimulation the control and SWAP-70 deficient cells showed no difference, upon stimulation with TGFb/EGF, only the control cells up-regulated their focal adhesion both in number and area. Thus, SWAP-70 is required for formation of focal adhesions in activated 4T1 cells.

Minor Points

1. The quantification of data shown in supplementary figure 2D, the colony forming assay, is provided in figure 2A. The supplemental figure 2D serves as an illustration of this experiment.
2. The general conditions used for RT-DC were in principle the same as for the AFM experiments, i.e. the cells were grown and stimulated in the same manner. Thus, the actin and myosin networks were initially the same. However, it is very difficult to investigate actin and myosin within an RT-DC experiments, i.e. during the very short milliseconds time period the suspended cells are squeezed through the analysis chamber. While one could have re-attached the cells to a surface just before putting them through the RT-DC to perform microscopical analysis of actin and myosin, this would be essentially useless since then one would again look at attached cells as was already done within the AFM set-up. Thus, we decided not do follow up on this.
3. This error is corrected, thank you for pointing it out.

May 6, 2024

RE: Life Science Alliance Manuscript #LSA-2023-02307R

Prof. Rolf Jessberger
TU Dresden
Inst. of Physiological Chemistry
Fiedlerstr. 42
Dresden 1307
Germany

Dear Dr. Jessberger,

Thank you for submitting your revised manuscript entitled "The F-actin bundler SWAP-70 promotes tumor metastasis". We would be happy to publish your paper in Life Science Alliance pending final revisions necessary to meet our formatting guidelines.

- please make sure the author order in your manuscript and our system match
- please place the Reference list before the figure legends
- please place the Supplemental Figure legends right after the main figure legends in the main manuscript file, rather than in a separate file
- please add an Author Contributions section to your main manuscript text
- please add a Conflict of Interest statement to your main manuscript text
- please use the [10 author names, et al.] format in your references (i.e. limit the author names to the first 10)
- please add the Twitter handle of your host institute/organization as well as your own or/and one of the authors in our system

FIGURE CHECKS

- for Figure S1, there are 2 "E" panels described in the figure legend, please update the second one to "F"
- please add scale bars to Figure S4E
- please add sizes next to blots in Figure S1E

A. FINAL FILES:

B. MANUSCRIPT ORGANIZATION AND FORMATTING:

Sincerely,

May 9, 2024

RE: Life Science Alliance Manuscript #LSA-2023-02307RR

Prof. Rolf Jessberger
TU Dresden
Inst. of Physiological Chemistry
Fiedlerstr. 42
Dresden 1307
Germany

Dear Dr. Jessberger,

Thank you for submitting your Research Article entitled "The F-actin bundler SWAP-70 promotes tumor metastasis". It is a pleasure to let you know that your manuscript is now accepted for publication in Life Science Alliance. Congratulations on this interesting work.

DISTRIBUTION OF MATERIALS:

Again, congratulations on a very nice paper. I hope you found the review process to be constructive and are pleased with how the manuscript was handled editorially. We look forward to future exciting submissions from your lab.

Sincerely,
